# Nature portrayed in images in Dutch Brazil: Tracing the sources of the plant woodcuts in the *Historia Naturalis Brasiliae* (1648)

**Mireia Alcantara-Rodriguez**[1]*, **Tinde Van Andel**[2,3], **Mariana Françozo**[4]

**1** Faculty of Archaeology, Leiden University, Leiden, The Netherlands, **2** Clusius Chair in History of Botany and Gardens, Institute for Biology, Leiden University, Leiden, The Netherlands, **3** Naturalis Biodiversity Center, Leiden, The Netherlands, **4** Associate Professor in Museum Studies, PI ERC BRASILIAE Project, Faculty of Archaeology, Leiden University, Leiden, The Netherlands

* m.alcantara.rodriguez@gmail.com

**Data Availability Statement:** Initially, all relevant data are within the manuscript and its Supporting Information files. Although I initially added them in

## Abstract

By the mid-seventeenth century, images of natural elements that originated in Dutch Brazil circulated in Europe. These were often included in art collections (the *Libri Picturati*) and natural history treatises (the *Historia Naturalis Brasiliae* and the *India Utriesque re Naturale et Medica*, 1658). The plant woodcut images in these books constituted (icono) type specimens and played a significant role in disseminating scientific botanical knowledge. We present a systematic analysis of their origins by cross-referencing the visual and textual sources related to Dutch Brazil. To do so, we used our previous botanical identifications of the portrayed plants, published sources, and digital archival material. The plant woodcuts accounted for 529 images, which corresponded to 426 taxa. We created a PDF booklet to visualize the (dis-) similarities of the woodcuts with the *Libri Picturati* and other visual sources. Substantial differences in the visual-making methodology exist between the two treatises (1648, 1658). In the first book, most of the images were available from Dutch Brazil and carved into the woodcuts, while most of these woodcuts were reused in the second one. The Indigenous Tupi-based plant names accompanying the images were crucial when arranging the sources, and portraying as much botanical information as possible was commonly the goal. Freshly picked, living plants, dried branches, fruits, and seeds were used to represent the megadiverse Brazilian flora, even when these belonged to species originating from other regions. Despite not being recognized for their contribution, Indigenous Brazilians and enslaved Africans were essential in the visual knowledge-making processes that later resulted in these natural history collections. As several sources remain lost and many histories yet untold, further archival studies and collaborative projects are pertinent to reveal the missing pieces of this conundrum.

the submission, a more quality copy of S2 and S4 (less compressed) is uploaded as a draft in the repository DANS Easy and these are available from https://doi.org/10.17026/dans-zk4-ercv (for S2) and https://doi.org/10.17026/dans-xm2-bnhw (for S4)

**Funding:** This study was funded by the European Research Council (ERC) under the Horizon 2020 Research and Innovation Program (Agreement No. 715423), ERC Project BRASILIAE: Indigenous Knowledge in the Making of Science, directed by Dr. M. Françozo at Leiden University. The funders had no role in study design, data collection and analysis, decision to publish, or preparation of the manuscript.

**Competing interests:** The authors have declared that no competing interests exist.

# 1 Introduction

## 1.1 Nature portrayed in the early modern period: Dutch Brazil

By the mid-seventeenth century, images of natural elements that originated in Dutch Brazil circulated in Europe, often included in art collections and natural history treatises. The term "Dutch Brazil" corresponds to the northeastern part of Brazil colonized by the Dutch West Indian Company (WIC) between 1630 and 1654 after they overtook it from the Portuguese. Appointed by the WIC, Count Johan Maurits of Nassau-Siegen was the governor-general in Dutch Brazil from 1636 to 1644. He commissioned naturalist George Marcgrave, physician Willem Piso, and painters Albert Eckhout and Frans Post, among others, to portray and document the environment encountered in the seized land. With the support of the WIC, the count commanded a military guard to accompany Marcgrave when collecting fauna and flora [1]. At the same time, local people (often Indigenous Brazilians, but also enslaved Africans and Portuguese) brought the specimens to him while he resided in Maurits' court. The same applied to Piso, who, like the other members of Johan Maurits' entourage, was entitled to an assistant for himself. The outcomes of the work done by the team assembled by Johan Maurits included, among others, the treatises *Historia Naturalis Brasiliae* (HNB) [2] and *India Utriesque re Naturale et Medica* (IURNM) [3], botanical specimens in Marcgrave's herbarium, and a set of plant illustrations, drawings, and sketches divided into three collections: *Theatrum Rerum Naturalium Brasiliae* (*Theatrum*), *Miscellanea Cleyeri* (*Misc. Cleyeri*), and the *Libri Principis* (also known as *Handbooks* or *Manuais*). The botanical imagery in these materials is the object of this study. By linking the images and collections from Dutch Brazil to each other and other contemporaneous works, we aim to trace the particular history of the images in the HNB as well as to shed light on how natural history books were produced in the early modern period.

## 1.2 Background to the visual and textual repertoire of Dutch Brazil

The origin, arrangement, and destination of all these materials are varied. The HNB was commissioned by Johan Maurits after his return from Brazil when he handed over the field notes of Piso and Marcgrave and several drawings on plants and animals to Johannes de Laet, WIC director, cartographer, and ultimately editor of the HNB. De Laet organized these drawings and ordered a few woodcut images based on Marcgrave's herbarium, as evidenced by his commentaries [4]. The HNB was published in 1648 by Elzevier in Amsterdam and Hackium in Leiden, becoming an authoritative work on Brazilian and tropical flora and fauna for the upcoming centuries [5, 6]. It contains references to classical naturalists, such as Dioscorides and Theophrastus, and to Renaissance and contemporary scholars (e.g., Francisco Hernández, Nicolas Monardes, and Carolus Clusius), whose manuscripts and early images of American flora influenced the work of Marcgrave and Piso [6, 7].

Based on the field notes, De Laet produced a preliminary draft of the HNB [8], which included 26 plant images (15 drawings and 11 proof woodcuts) and 366 plant descriptions–often with the word *Icon* next to them [5]. This manuscript could be De Laet's first attempt to arrange Marcgrave's species. The *Icon* term could correspond to field drawings, watercolor illustrations, oil paintings from the expedition [9], or living plants collected in Brazil [5]. The manuscript was later purchased by the slaveholder, physician, and collector Hans Sloane (1660–1753), and it is now part of the Sloane manuscript collection at the British Library. Most of Marcgrave's specimens ended up in Copenhagen in 1653 and are now kept at the University Herbarium (C).

In 1644, Johan Maurits, Piso, Eckhout, Post, and other members of the count's entourage returned to the Low Countries, but Marcgrave never did. Instead, in 1643, he was sent to

Angola, where he died shortly after his arrival [10]. De Laet passed away in 1649, a year after the publication of the HNB. Discontent with the HNB arrangement, Piso edited its content and published a modified version in 1658 [3].

The Brazilian imagery had another destiny. In 1652, Johan Maurits sent several oil-based illustrations and drawings of flora, fauna, and Indigenous and African peoples from Dutch Brazil as a diplomatic gift to Frederick William, Elector of Brandenburg from 1640 to 1688. He passed them to his court physician, Christian Mentzel, who organized the oil paintings in a bound collection (the *Theatrum*). Mentzel included Johan Maurits's words in the preface of the *Theatrum*, emphasizing that "these images aimed to reproduce Brazilian nature as perfectly as possible" [11: 18]. The elector also received two bound volumes of watercolors (the *Libri Principis*), and a few sketches and oil paintings, which were bound in 1757 (the *Misc. Cleyeri*) [5]. In the nineteenth century, these Brazilian collections were incorporated into a more extensive collection known as the *Libri Picturati*, which was housed in the Preussische Staatsbibliothek (the present-day *Staatsbibliothek* in Berlin, Germany) until this library was evacuated during World War II [12]. The *Libri Picturati* were considered lost until zoologist Peter Whitehead located them at the Jagiellonian Library in Krakow, Poland [5].

## 1.3 Behind the images: Previous and present research

Due to its undeniable historical value, several studies on the Brazilian images in the *Libri Picturati* have analyzed their content, authorship, and connection to Marcgrave and Piso's works. Scholars have argued that the *Libri Picturati* images served as models for the woodcuts of the HNB [5, 12–20, 22–25]. When the collection was still in Berlin, botanist Carl Friedrich Philipp von Martius [13] studied Marcgrave and Piso's plants and noticed some species were included in the oil paintings. One example is *Eugenia uniflora* L. (known as *Manaca*), the woodcut of which in the HNB was presumably made after two images illustrated in the *Theatrum*: "ubi icones duae, e quibus xylographica in libro composita" (13: 5). Zoologists Hinrich Lichtenstein [14, 15] and Anton Schneider [16] matched several of the animal illustrations in the *Theatrum* with the woodcut images in the HNB. While the *Libri Picturati* remained still "lost," Whitehead [12, 17] (who suspected it was being kept in Poland) brought the scholarship attention to the relevance of connecting the visual material of Dutch Brazil to the textual sources [2, 3]. After rediscovering the collection, the zoologist and his colleague Marinus Boeseman [5] attributed some parallels between the woodcuts and the *Libri Picturati* images, focusing mainly on the animals [18]. They proposed the existence of designs made before the oil paintings that had acted as models for both works. Likewise, art historian Petronella Albertin [19] mentioned the oil paintings as models to elaborate the woodcuts. A decade later, Dante Teixeira [20] found several similarities between the *Libri Picturati* and the woodcuts, especially for the animals, and he also recommended studying the visual and textual material as a whole to understand the nucleus of models from which it originated.

In our previous research, we identified the plant species depicted in the whole Brazilian collection within the *Libri Picturati* and found that a substantial number of species were documented and often illustrated in the HNB and IURNM [21]. However, to what extent the images representing these species were comparable between the *Libri Picturati* and the HNB was still a mystery, as well as the potential authors behind their making. Recently, Rebecca Brienen [22] attributed most of the *Theatrum* plant illustrations to Eckhout, as Thomas Thomsen (biographer of Eckhout) had done when the collection was still in Berlin [23]. In recent years, visual art researcher Cláudia Scharf [24] compared the watercolors of the animals in the *Libri Principis* with the HNB woodcuts and found remarkable similarities among them, attributing its authorship to Marcgrave and inferring the authorship of the flora woodcuts to

Eckhout based on Brienen's analysis [22]. According to Wolfgang Joost [25], a few woodcuts in the HNB were based on Post's drawings. Brazilian botanists José Pickel [7] and José de Almeida [6] suggested De Laet "borrowed" some images from published treatises, such as those by Clusius or from De Laet's work on Amerindian flora [26, 27]. Some woodcuts were made in the Dutch Republic after the specimens collected by Marcgrave [9], and several were based on Marcgrave's drawings, as De Laet and Piso indicated in the preface of the HNB [2] and the IURNM [3]. In 1640, Marcgrave wrote to De Laet from Brazil that he had made 350 pencil drawings of plants and several more of animals [28]. The few pencil sketches found in De Laet's manuscript could be part of those [5, 9], but most of Marcgrave's original drawings do not exist anymore or have not been located yet.

Beyond doubt, the study of the woodcuts in these natural history books is of great relevance, as they play a significant role in the transmission of scientific botanical knowledge [29]. More-over, the woodcuts constituted (icono) type specimens because they accompanied the first descriptions of individual species against which all later individuals were compared [5]. As the intriguing question of which plant images were used as the basis for the woodcuts remained unanswered, we present here a systematic analysis of their origins. We focused on the botanical images included in the HNB, the IURNM, the *Libri Picturati* Brazilian collection, De Laet's manuscript, and Marcgrave's herbarium specimens to 1) analyze the (dis-) similarities between the woodcuts and the other visual sources from Dutch Brazil and 2) trace back the remaining sources that were used to create the woodcuts. By applying our botanical image analysis to these historical collections, we provide an overview of how the visual material was used in the composition of seventeenth-century natural history treatises on Brazil, and we add insights into the processes of visual knowledge-making and botanical practices in the early modern period.

## 2 Materials and methods

To analyze the (dis-) similarities among the historical visual sources, we built a database in FileMaker Pro with all woodcuts and illustrations organized by species and created a spread-sheet with the background information. Our database contained all digital woodcut images in Marcgrave's and Piso's books [2, 3], their correspondent images for the same species in Marc-grave's herbarium (collected between 1638 and 1643), the *Libri Picturati*, and other visual sources. The surviving copies of the HNB amount to 302, distributed in collections and institutions worldwide [30]. We used the digital-colored copy of the HNB located in Leiden University Library in the Netherlands [Shelfmark 1407 B3 [30]] (HNB Leiden Universiteitsbibliotheek) (Fig 1) and the digital images of Marcgrave's herbarium in C (Marcgrave's herbarium). For the IURNM, we used the copy kept at the Missouri Botanical Garden (IURNM copy Missouri). The Jagiellonian library provided the *Libri Picturati* illustrations as digital images, of which the *Theatrum Rerum Naturalium* (Theatrum), *Miscellanea Cleyeri* (Misc. Cleyeri) and the *Libri Principis* (Libri Principis) are now publicly available.

To trace the origins of the woodcut images in the HNB and the IURNM, we also compared the woodcuts with the drawings included in De Laet's manuscript in the British Library's Sloane manuscript collection [8]. In addition, we associated the plant descriptions with the term "Icon" to their corresponding species in the HNB and the *Libri Picturati* to analyze potential matches between their accompanying vernacular plant names. The Manuscript Department of the British Library provided the digitized scans of De Laet's manuscript.

To compare woodcuts, vouchers, and plant illustrations, we cross-referenced all textual and visual sources for each plant species using our recent botanical identifications for the HNB, the IURNM, and the *Libri Picturati* [21, 31]. Then, we assembled the images from multiple sources belonging to the same species to visualize their (dis-)similarities using FileMaker Pro. Due to

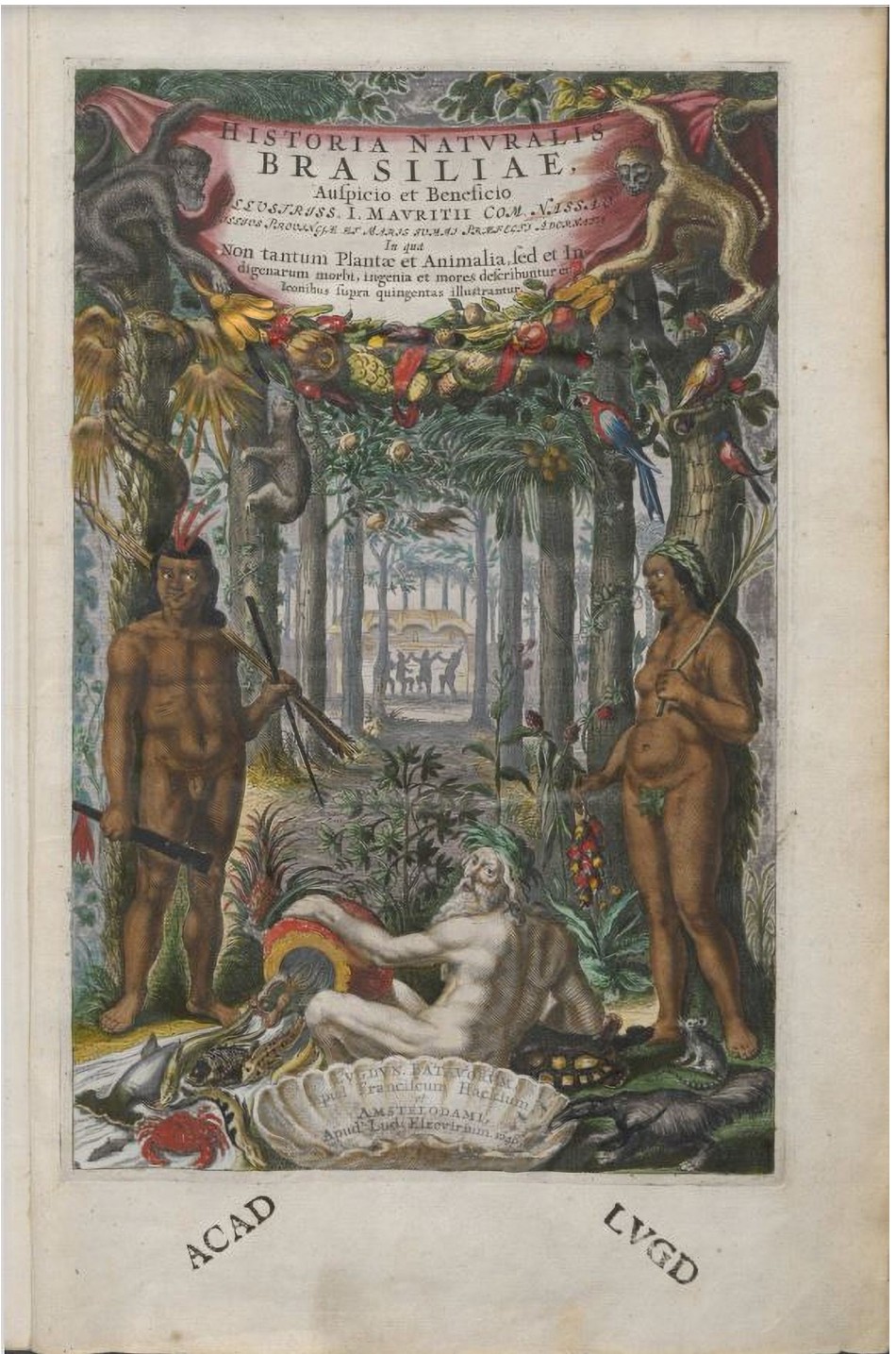

**Fig 1. Frontispiece of the *Historia Naturalis Brasiliae* (Marcgrave and Piso, 1648).** Colored copy kept at the Library of Leiden University (the Netherlands). Available from https://digitalcollections.universiteitleiden.nl/view/item/1535938#page/25/mode/1up.

the more extensive number of woodcuts in Marcgrave's chapters on plants [4] than in Piso's work [3, 32] and the many duplicates in the latter, we started our analysis with Marcgrave (1648). We systematized the information with database entries on the page numbers of the

woodcuts in the HNB and IURNM, vernacular name(s), botanical identifications, and the origin of these woodcuts.

We first analyzed the woodcuts that represent species in common with Marcgrave's herbarium and the *Libri Picturati*. For the latter, we identified different degrees of similarity based on their botanical elements (or the lack thereof), spatial composition (indicating when the images were similar but reversed), shape, and form. We distinguished four categories: 1) very similar (woodcut and illustration share [almost] the same elements, shape, and form), 2) moderately similar (they bear a remarkable resemblance, especially in their morphology, but do not share precisely the same elements, or these are placed differently), 3) slightly similar (they are different in most elements yet some, such as the inflorescence, the fruits, a few leaves, etc., look-alike), and 4) different (the images do not present any similarity in their composition and botanical elements). The more similar a woodcut and its corresponding species in the *Libri Picturati* were, the more probably the woodcut was made after the image, or they both originated from the same source. For the woodcut images that did not resemble the visual sources mentioned above, we checked the works of Renaissance and early modern scholars that included engravings similar to the plant images in the HNB and IURNM. To navigate this vast corpus of literature, we first checked the HNB and IURNM for references to scholars who worked with tropical flora, such as Hernández [33], Monardes [34], or Clusius [35–37]. We narrowed this search by consulting Almeida [6], who systematized those citations per plant species. For Clusius, we searched for corresponding images in Ubrizsy and Heniger [38], who listed the American plants portrayed by this botanist. For the woodcuts whose sources were still unknown, we searched for their species name on Plantillustrations.org (plantillustrations.org). This open-access site offers an extensive digital collection of HD botanical illustrations through time. We retrieved some of these flora illustrations for our database.

Finally, we checked phenological and other botanical characters in modern photos (including herbarium vouchers) and added them to the database to show how these plants appear in nature. We retrieved all plant images from Creative Commons (creativecommons.org), Flickr (flickr.com), Plants of the World Online (powo.science.kew.org), Flora do Brasil 2020 (floradobrasil.jbrj.gov.br), Species Link (specieslink.net), and the Global Biodiversity Information Facility (gbif.org).

## 3 Results

### 3.1 Origin of the *Historia Naturalis Brasiliae* (1648) woodcuts

There are 301 plant woodcuts in the HNB [2]. We listed the background information of all woodcut images (species, family, sources, page numbers, author, etc.) in S1 Data. We provided in S1 Appendix an overview of all the visual sources arranged by their correlated species, taking as the reference the woodcuts in the HNB. Fig 2 shows how we linked a woodcut in the HNB to several other visual sources.

Several woodcuts in Marcgrave's chapters on plants are repeated in Piso's chapter [39] because De Laet re-used them for the species mentioned by both authors (Fig 3). In the HNB, nine species are represented by two different woodcuts, and two species are depicted by three. For six woodcuts, we could not identify the species shown (Fig 3).

We traced approximately one-third (84 woodcuts) of the 243 unique woodcuts in the HNB (Marcgrave's woodcuts plus Piso's unique ones) to the *Libri Picturati*, Marcgrave's herbarium, or other known sources (Fig 3). However, over two-thirds (159 woodcuts) of the plant images did not correspond to these visual sources.

**3.1.1 *Theatrum Rerum Naturalium*.** From the entire Brazilian collection of the *Libri Picturati*, the HNB plant woodcuts mostly match the *Theatrum* illustrations, not the *Libri Principis*'

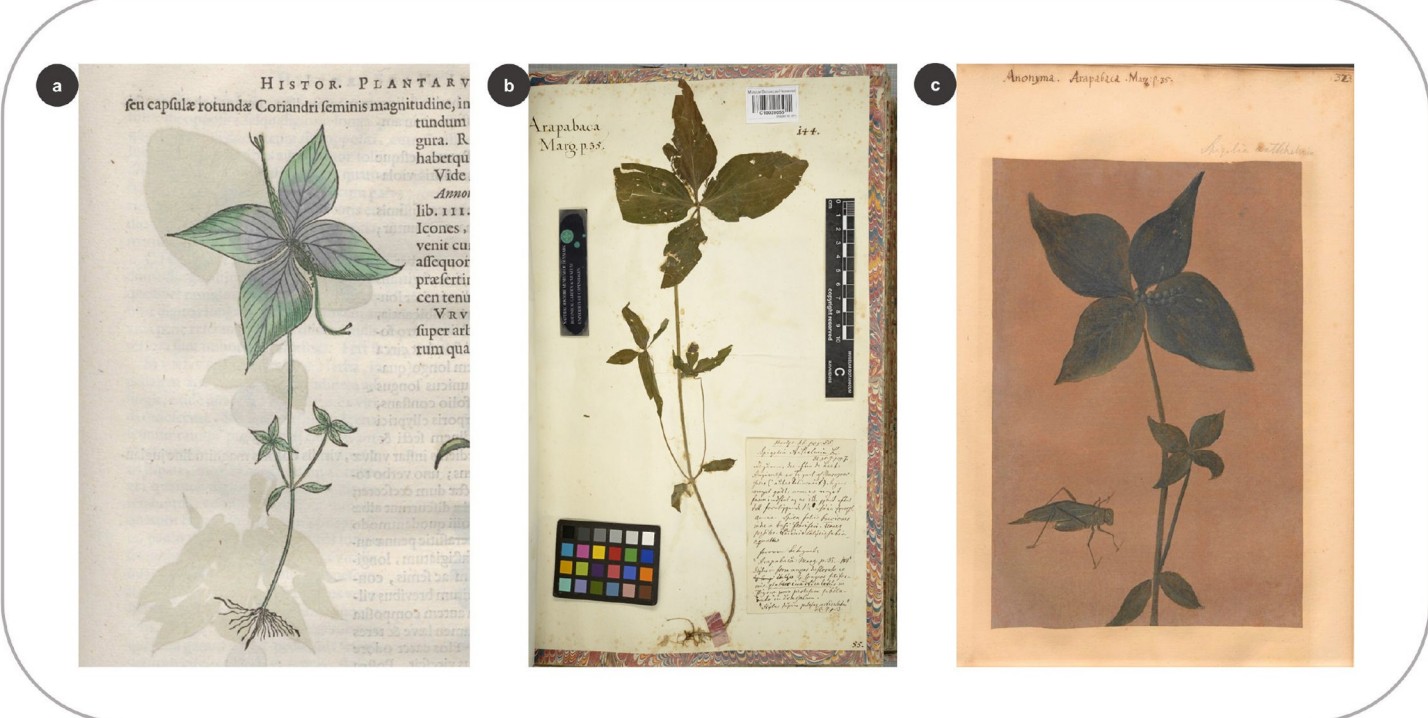

**Fig 2. Linking a woodcut in the HNB to other visual sources from Dutch Brazil.** (a) Woodcut of the common medicinal plant against intestinal worms, *Spighelia anthelmia* L. (Marcgrave 1648: 35) (b) Specimen of the same species in Marcgrave's herbarium (f. 55) (c) *S. anthelmia* depicted as an oil painting in the *Theatrum* (f. 323).

watercolors or the *Misc. Cleyeri's* drawings. One exception is the woodcut that represents *Canna indica* L., which is very similar to the painting in the *Misc. Cleyeri* (in reversed format). Surprisingly, most of the plant woodcuts in the HNB represent species not included in the *Theatrum* (Fig 3). Of those species in common (78 species in Marcgrave and 15 spp. in Piso), around half in Marcgrave and 60% in Piso bear resemblance in different degrees to the oil paintings (Figs 4 and 5). In addition, a small proportion (15%, seven spp.) of the overlapping images between the HNB and the *Libri Picturati* (47 spp.) are reversed (example in Fig 5B).

**3.1.2 Marcgrave's herbarium.** Of the 143 species preserved in Marcgrave's herbarium, 92 (64%) are described in Marcgrave and Piso's books, but only 74 are represented by a woodcut image in the HNB. According to Andrade-Lima et al. [9], 17 woodcuts for the HNB were produced using Marcgrave's herbarium vouchers (Fig 3 and S1 Data). De Laet explicitly mentioned in the HNB that 15 woodcuts were made after these specimens [4]. In addition, *Galphimia brasiliensis* (L.) A.Juss. and *Spondias mombin* L. were also made after Marcgrave's exsiccates (S1 Appendix: 303, 449), although De Laet did not indicate this in the HNB [9]. By comparing the taxa shared between the HNB and the herbarium, we found that the woodcuts of 25 species bear no resemblance to the vouchers; hence, these were probably made after other sources. For 17 species in the herbarium, we cannot infer whether the specimens were used to design the woodcuts because they are poorly preserved or consist of a few plant parts (a single leaf or sterile branches). Nevertheless, 17 specimens share similarities to the woodcut images (Fig 3). These herbarium vouchers could have been used as models to make the drawings later carved onto the woodblocks (Fig 6).

**3.1.3 Other traceable visual sources.** De Laet tried to find ways to provide images of the plants described in the HNB when these were lacking–as occurred with the vouchers collected

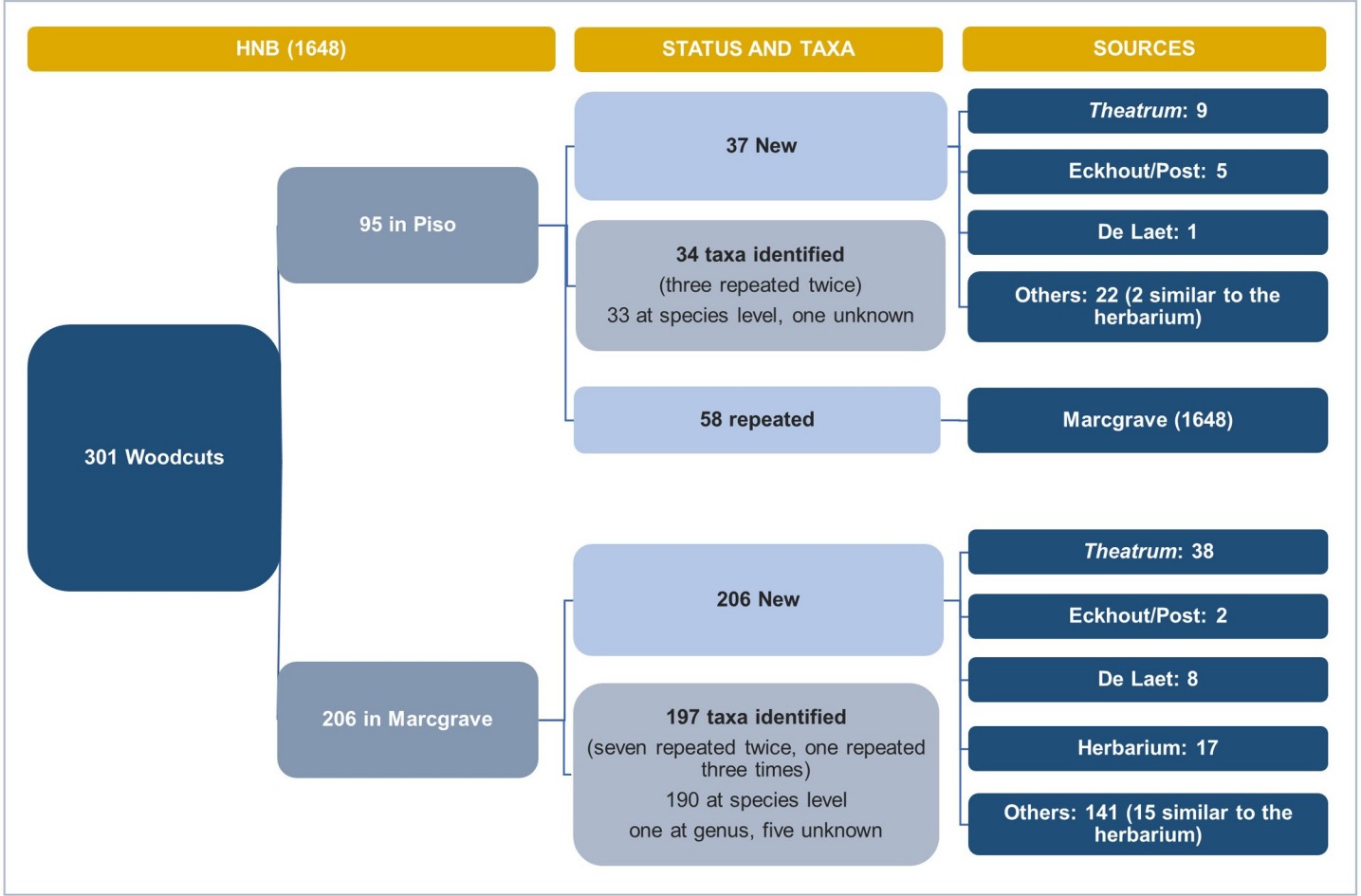

**Fig 3. Tracing the sources of the plant woodcuts in the HNB.** Flowchart showing the organization of the woodcut images in the HNB, the number of plant taxa, and the sources used to create the woodcuts.

by Marcgrave. For example, he received a pod of *Mimosa pigra* L. from his friends in Brazil. De Laet commissioned to have it designed at natural size, even though "the painter hardly represented its elegance" (4: 74). While editing the HNB, he also obtained seeds from Brazil of a plant called *Micambeangolensibus*. He planted them and observed how the plant grew and flowered in the summer of 1646 but saw it die in October [4]. The seeds belonged to *Cleome gynandra* L., an African weed introduced to Brazil via the trans-Atlantic slave trade and eaten by enslaved Africans as a leafy vegetable (4: 10). De Laet tried to obtain a drawing from the living plant in the Dutch Republic. Yet, he did not manage to do so, as the tropical plant perished during the winter frost [4].

We found that eight woodcuts were based on images previously published by De Laet in his books on the colonized territories of the Americas [26, 27]. For instance, De Laet mentioned that the woodcut of *Bromelia karatas* L. depicted on p. 111 in Piso's chapter on medicinal plants [39] was drawn at its natural scale after he received the fruit from Brazil (26: 614). In turn, De Laet (re)used three of the woodcuts from his books, which he copied after images of related flora published by Clusius (364,375) (S1 Appendix). He often cited Clusius, Monardes, and other Renaissance authors to compare the Brazilian plants to known European or American species.

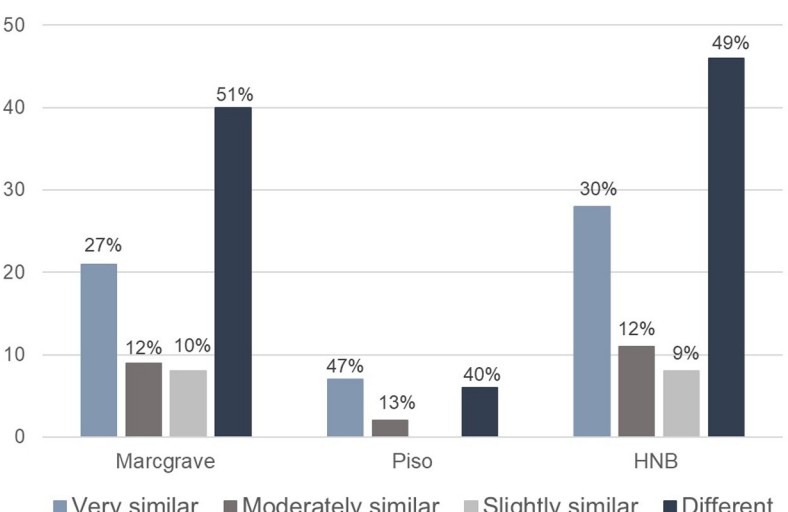

**Fig 4. Comparing the plant images portrayed in the HNB and the *Theatrum* for the same species.** Similarities between the botanical illustrations in the *Theatrum* and the woodcuts depicted in Marcgrave's chapters on plants, Piso's chapter on medicinal plants, and the whole repertoire of botanical woodcuts in the HNB (1648).

The artists who joined Johan Maurits's crew in Brazil also played a role in the making of the images. Three woodcuts included in Piso [39] depict how sugarcane (*Saccharum officinarum* L.) and cassava (*Manihot esculenta* Crantz) were processed in the mills and ovens of the colony by enslaved African labor. Some scholars attributed these images to Frans Post [5, 25]. We found similar scenes in Barlaeus' 1647 publication about Johan Maurits' endeavors in the colony [40], which Post indeed designed and which appear embedded in the maps Marcgrave drew in Brazil (S1 Appendix: 1–5). The woodcut of *Cereus jamacaru* DC. also closely resembles the cactus portrayed by Post in "View of the Rio São Francisco Brazil with Fort Maurits and Capibara" (S1 Appendix: 437).

Post's contemporary Albert Eckhout included at least three plants in his still-life paintings and portraits similar to the woodcuts in the HNB (S1 Data and S1 Appendix). One example is *Ipomoea pes-caprae* (L.) R. Br, of which the woodcut in Marcgrave's chapter in the HNB does not match any of the known sources (Fig 7A), but the same species in Piso (Fig 7B) strongly resembles the creeping plant in Eckhout's portrait known as "African man" (Fig 7C and 7D). There is a crayon sketch of the same species in the *Misc. Cleyeri* (Fig 7E), but it neither resembles the woodcut (Fig 7B) nor the vine in the painting (Fig 7D)–as is often the case with the sketches in the *Misc. Cleyeri* and Eckhout's paintings [22, 23].

**3.1.4 Remaining sources of the HNB woodcuts.** Most of the HNB images (67%, 163 woodcuts) did not originate after the *Libri Picturati*, Marcgrave's herbarium, Eckhout and Post's paintings, or De Laet. The question is whether Marcgrave himself made the original drawings. Interestingly, considerable mistakes are present in several of these woodcuts. As Pickel [7] noticed, the banana bunch (*Musa × paradisiaca* L.) emerges from the middle part of the trunk instead of the center (4: 137). The *Anda* tree (*Joannesia princeps* Vell.) bears large, bell-shaped flowers (4: 110) compared to the smaller and compound inflorescences characteristic of the Euphorbiaceae family (S1 Appendix: 377–378). We agree with Pickel that a trained botanist like Marcgrave would hardly have approved these morphological errors in his botanical drawings.

In Marcgrave's chapter on trees, De Laet indicated for *Jetaiba* (*Hymenaea* cf. *courbaril* L.) that "more details about this tree were to be found in Piso, to whom we owe this figure" (4:

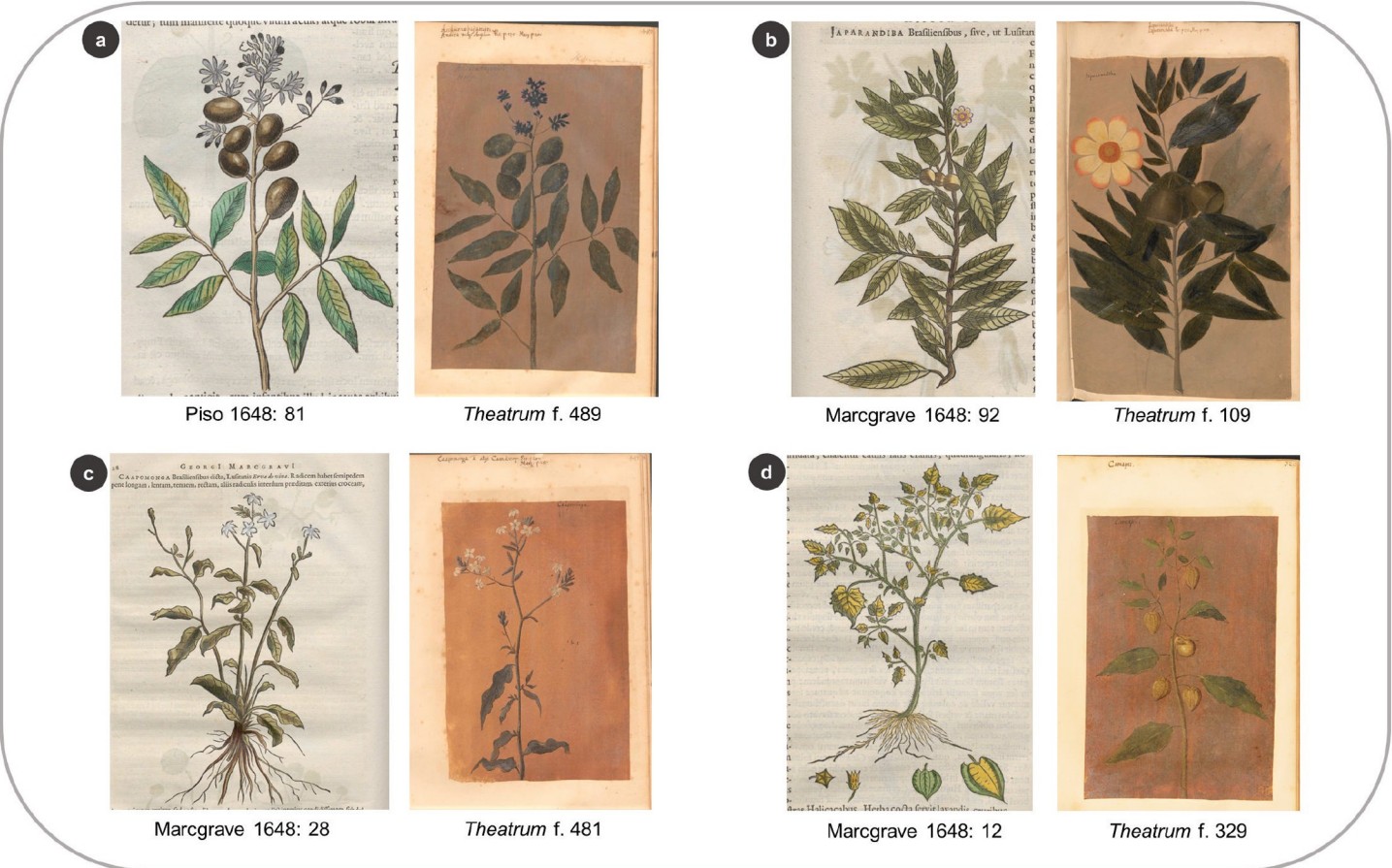

**Fig 5. Examples of similarity degrees between the woodcuts in the HNB and the *Theatrum* illustrations.** (a) Very similar: *Andira fraxinifolia* Benth. (b) Moderately similar: *Gustavia augusta* L. (c) Slightly similar: *Plumbago zeylanica* L. (d) Different: *Physalis pubescens* L.

101). De Laet referred to the woodcut in Piso, which is the same as in Marcgrave's chapter (39: 60). It is unclear whether Piso made some of the plant drawings since he was not trained in botany and art like his naturalist colleague. We know that some woodcuts were borrowed from Marcgrave's sketches during their expeditions [3: 107, 41: 249], while others were made *ad vivum* (after-life) by a painter traveling with him in the *sertão*, the dry hinterland of northeastern Brazil (preface in [3: 2, 41: 8]).

### 3.2 Origin of the woodcuts in the *India Utriusque re Naturale et Medica* (1658)

We listed the background information of all woodcut images (species, family, sources, page numbers, author, etc.) in S2 Data and displayed their associated images in S2 Appendix. Piso reused many woodcuts from the HNB, which we showed in S1 Appendix. Hence, to avoid repetitions, we only added in S2 Appendix those different plant woodcuts. In most cases, the woodcuts from Marcgrave's chapters on plant species he did not describe as used by humans are lacking in the IURNM.

**3.2.1 New and reused woodcut images.** In the IURNM, there are 228 woodcuts, eight of which are depicted twice (Fig 8). Hence, Piso used 220 woodblocks to complete the botanical part of his *solo* work. He distributed the images in Chapter IV (equivalent to Marcgrave's

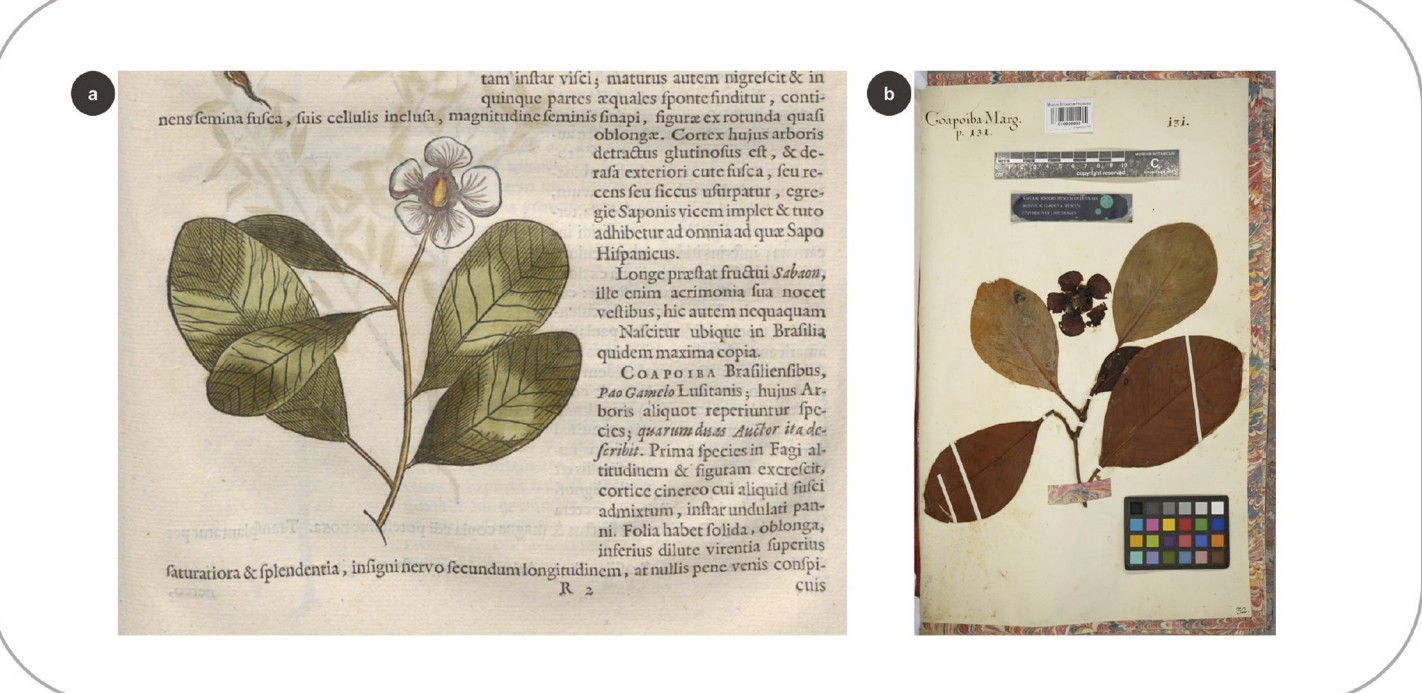

**Fig 6. Matching elements between a woodcut in the HNB and the specimen collected by Marcgrave.** (a) Woodcut of *Clusia nemorosa* G.Mey. in the HNB (Marcgrave 1648: 131) (b) Specimen of *C. nemorosa* in Marcgrave's herbarium (f. 32).

chapters on flora and Piso's chapter on medicinal plants in the HNB [42]) and Chapter V (equal to his chapter on venoms and antidotes in the HNB [43]).

Within the 59 identified taxa (Fig 8), we found a mushroom, a sponge, and a coral, which do not belong to the plant kingdom but were classified as such by seventeenth-century scholars. Most of the woodcuts (72%) look exactly like those printed before in the HNB, suggesting these were made with the same woodblocks De Laet had used ten years earlier, which were kept by the Elzevier publishing house after his death [5]. The remaining woodcuts (28%) were "new" illustrations Piso included in the IURNM. The physician copied most of those images (40%) after Renaissance and Early Modern botanists and physicians' herbals, such as those written by Brunfels [44], Fuchs [45], Matthioli [46], and Monardes [34, 47] (S2 Data and S2 Appendix). He also "borrowed" several images from the works of Dodoens [48, 49], De l' Obel [50, 51], and Clusius [35–37]. Sometimes, Piso did not depict plants collected or observed in Brazil, like *Gossypium barbadense* L. and *Astrocaryum vulgare* Mart. Instead, he copied the African cotton (*Gossypium arboretum* L.) and the date palm (*Phoenix dactylifera* L.) from the work of Prospero Alpini [52, 53]–a Venetian physician who depicted the flora he encountered during his expedition in Egypt.

Several of those new woodcuts (28%) are modified copies made after the images in the HNB related to the same plant. Most of these modifications consisted of attaching the branch depicted in the HNB to a trunk–a style often encountered in late Renaissance herbals, likely to portray the tree habit of the plant while still showing enough detail of the fertile branch. Occasionally, some new images were created by combining parts of different plant species, something that Pickel (7: 23) defined as "fantasy woodcuts" (S2 Data and S2 Appendix: 29, 85, 87, 103). A few times, the modified plant parts in the woodcut were those that had nutritional or

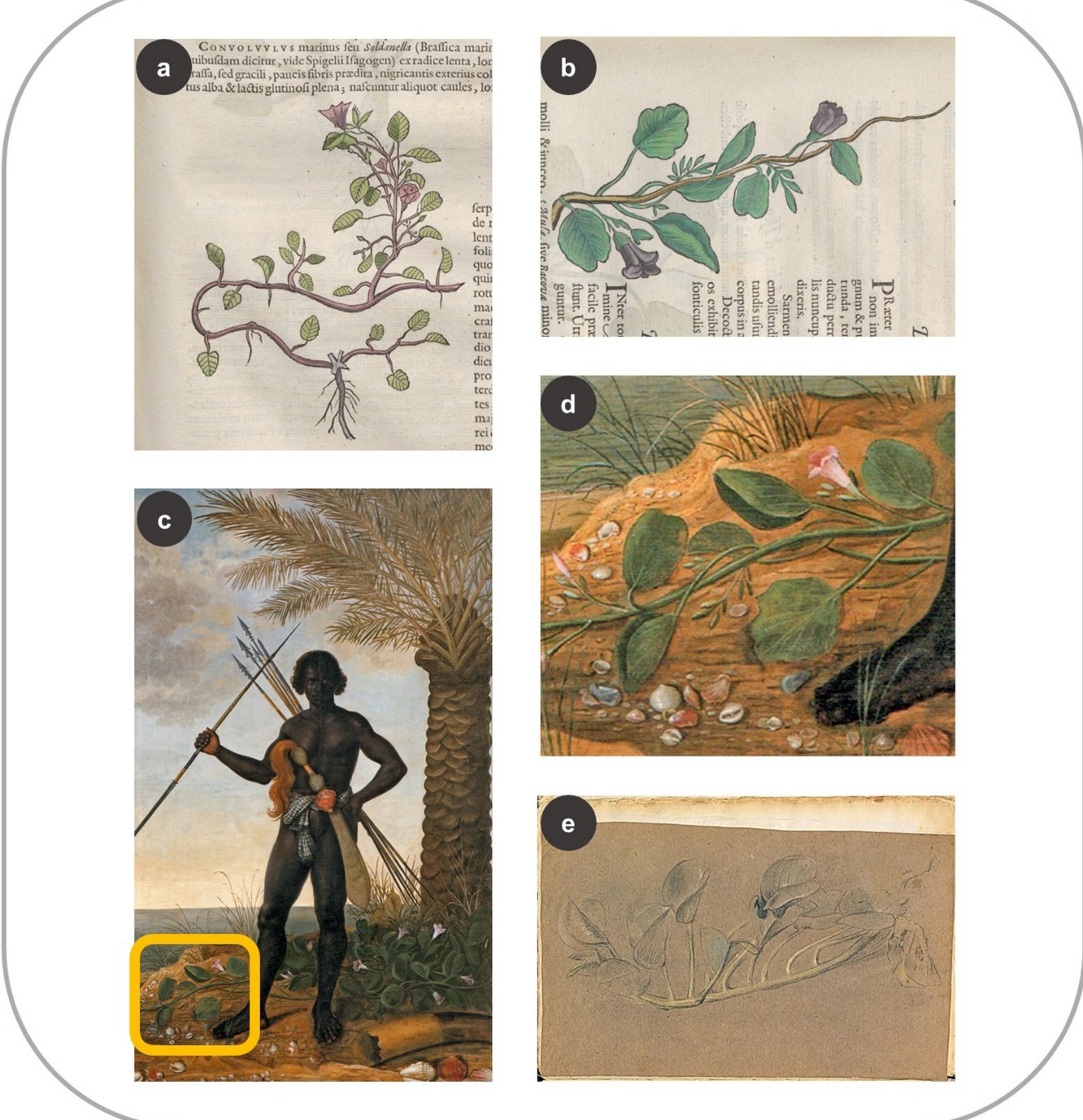

**Fig 7. Similarities between the HNB and Eckhout's paintings.** (a) Woodcut of *Ipomoea pes-caprae* (Marcgrave 1648: 51) (b) *I. pes-caprae* in Piso (1648: 103) (c) the same species on the left bottom corner in Eckhout's painting (National Museum of Denmark) (d) close-up of *I. pes-caprae* in Eckhout's painting (e) Sketch of the same species in the *Misc. Cleyeri* (f. 12v).

medicinal value. For example, two woodcuts represent *Cissampelos glaberrima* St. Hil. (*Caapeba*) in the IURNM (Fig 9A). One is the same as in the HNB (Fig 9B) and was made after one of the woodblocks that remained with Elzevier's publishers. The other woodcut is new, albeit slightly similar to the first one, with bigger roots that split in two (Fig 9A). Piso (42: 261)

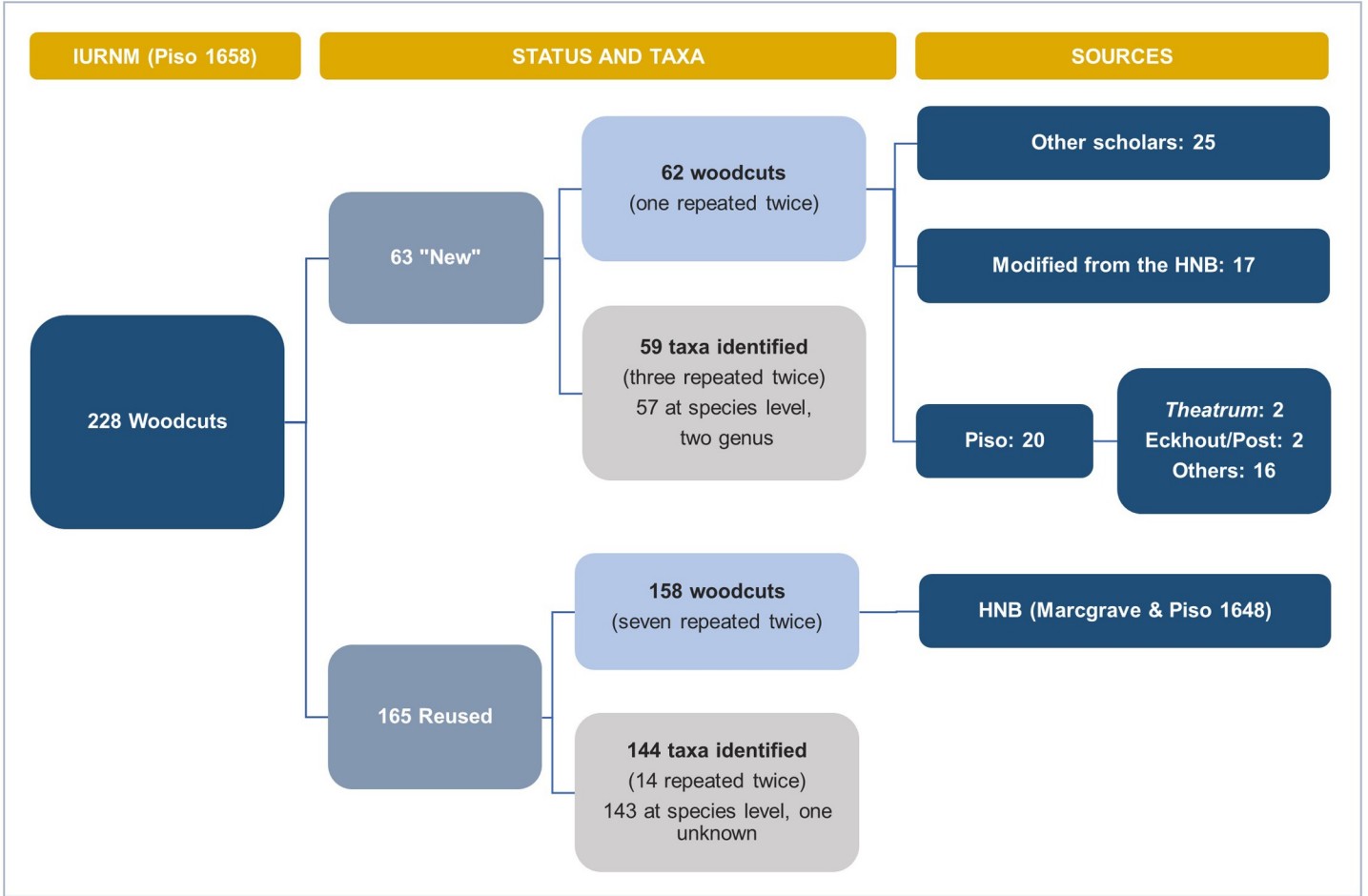

**Fig 8. Tracing the sources of the plant woodcuts in the IURNM.** Flowchart showing the organization of the woodcut images in the IURNM, the number of plant taxa, and the sources used to create these woodcuts.

directed the reader to these different roots by citing this figure, specifying that one of them became larger as it grew older. In addition, he experimented with the leaves and roots of *C. glaberrima* and documented its medicinal properties [42].

Three years before the publication of the IURNM, Danish physician and historian Ole Worm published an image of *C. glaberrima* root in his book *Museum Wormiamum* [54]. Worm obtained this root from Brazil (Fig 9C), perhaps as part of the plant material he exchanged with De Laet [9]. He also added the woodcut from Marcgrave using the same woodblock De Laet applied in the HNB (Fig 9D), owned by their common publisher: Elzevier [5, 9]. Just like Worm, Piso was aware of the medical relevance of *C. glaberrima* and emphasized its roots in his new image.

The remaining new woodcuts (32%) were not copied after the HNB woodcuts or images from other treatises. Four resemble the illustrations in the *Theatrum* (S2 Data and S2 Appendix: pp. 23, 57) and the flora depicted by Eckhout or Post in their paintings (S2 Data and S2 Appendix: 1, 39). Most of them (15 woodcuts), we could not trace to any existent source. Piso could have had sketches and seeds of the plants on those images, but their exact origin or provenance remains uncertain. Sometimes, the plant drawings did not make it. For instance, the physician did not provide the woodcut of *Cuipouna* (*Cestrum schlechtendalii* G.

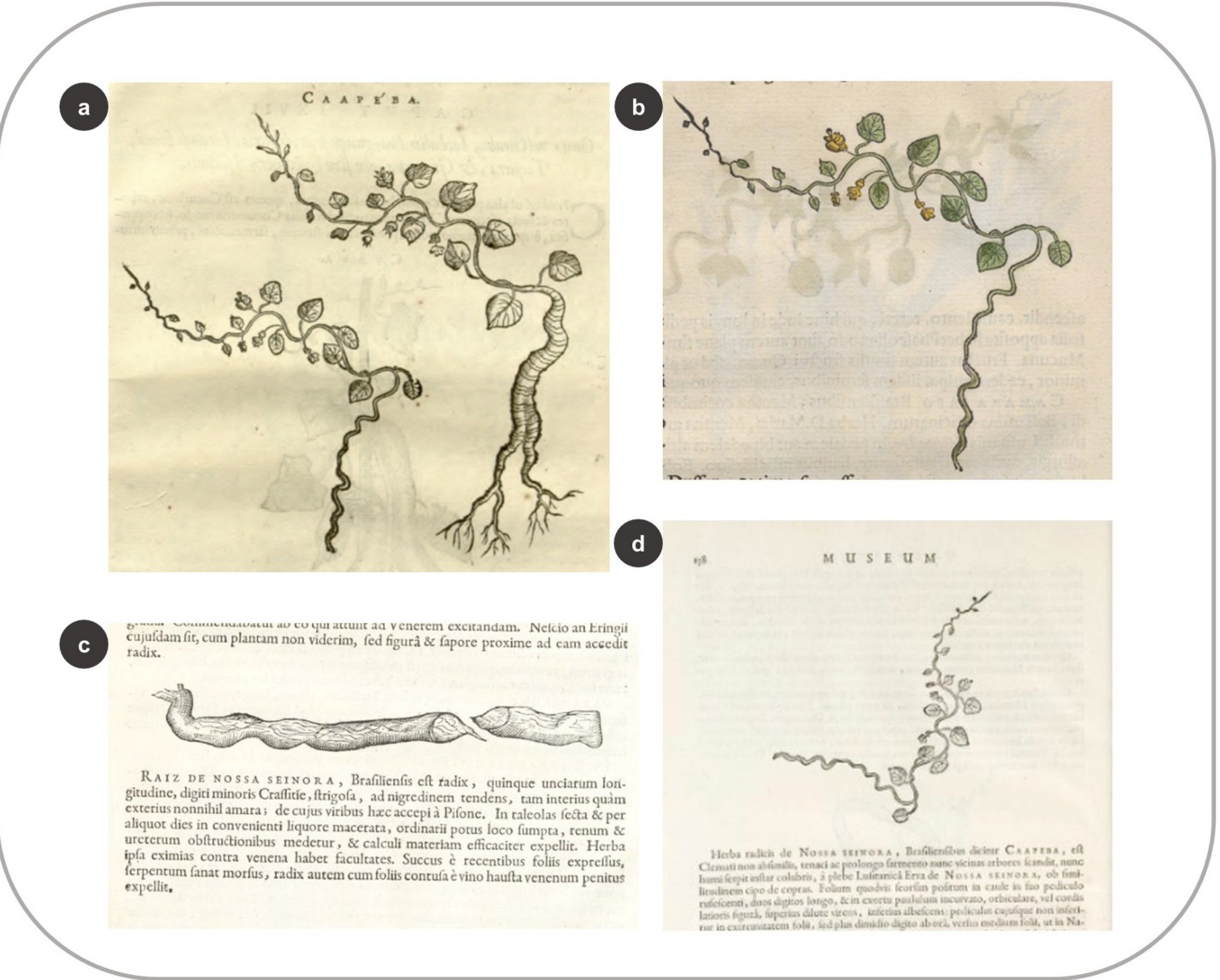

**Fig 9. New image used by Piso (1658) and its connection to previously published woodcuts.** (a) Woodcuts of *Cissampelos glaberrima* St. Hil. In the IURNM (page 261) (b) The same species in the HNB, documented by Marcgrave (p. 26) (c) Root of *C. glaberrima* shipped from Brazil as portrayed by Worm (1655: 157) (d) The same species made after the same woodblock (Worm 1655: 158).

Don) because its picture "had been damaged by the action of time" (42: 178). He lost the drawing of *Tapirapecu* (*Elephantopus mollis* Kunth) due to the "eventualities of the journey" (42: 182). These circumstances may refer to expeditions in Brazil or events happening during his return to the Low Countries or after it.

Of the 143 species preserved in Marcgrave's herbarium, 55 are represented by a woodcut image in the IURNM. However, only seven species from the herbarium appear in the newly made woodcuts and not in the HNB. There is no resemblance between Marcgrave's herbarium vouchers and their corresponding species in the new IURNM woodcuts.

### 3.3 Early modern "Image editing"

**3.3.1 Combining multiple sources.** Grouping images from various sources was a technique that allowed portraying a plant as completely as possible, especially when it was challenging to capture at once the different parts of the plant or when some details were missing. A woodcut in the HNB represents the medicinal shrub *Jatropha curcas* L. (Fig 10A), with lesser quality and scientific detail than its homolog in the *Theatrum* (Fig 10B). Despite the poor quality, De Laet included this image instead of making a new woodcut after the *Theatrum* oil painting. He either did not see this painting or deliberately chose the drawing of lesser quality to avoid making a new woodcut design, thus saving time and money. The HNB woodcut was presumably created after a drawing by Marcgrave (7: 129). Ten years later, Piso published a different image in which the internodal scars on the stem are visible (Fig 10C). This new image is slightly similar to the one in the *Theatrum*, although the ring-like scars on the stem are not visible in the oil painting. Marcgrave documented this feature on p. 96 and compared it to a fig tree, which might explain why Piso added it to the image. Additionally, the seeds of *J. curcas* are displayed in both the HNB and the IURNM (Fig 10C and 10D). This image was made "au naturel" after the seeds that De Laet (27: 137) received from Brazil and used in his treatise on the Americas (Fig 10E).

Another example of using multiple sources occurs with *Bixa orellana* L., known today as *Achiote*, *Annatto*, or *Urucu(m)* in Brazil–among others (Dataplamt, accessed 23.09.22). The woodcut in the HNB shows a flowering branch with four terminal fruits and small fruits on a

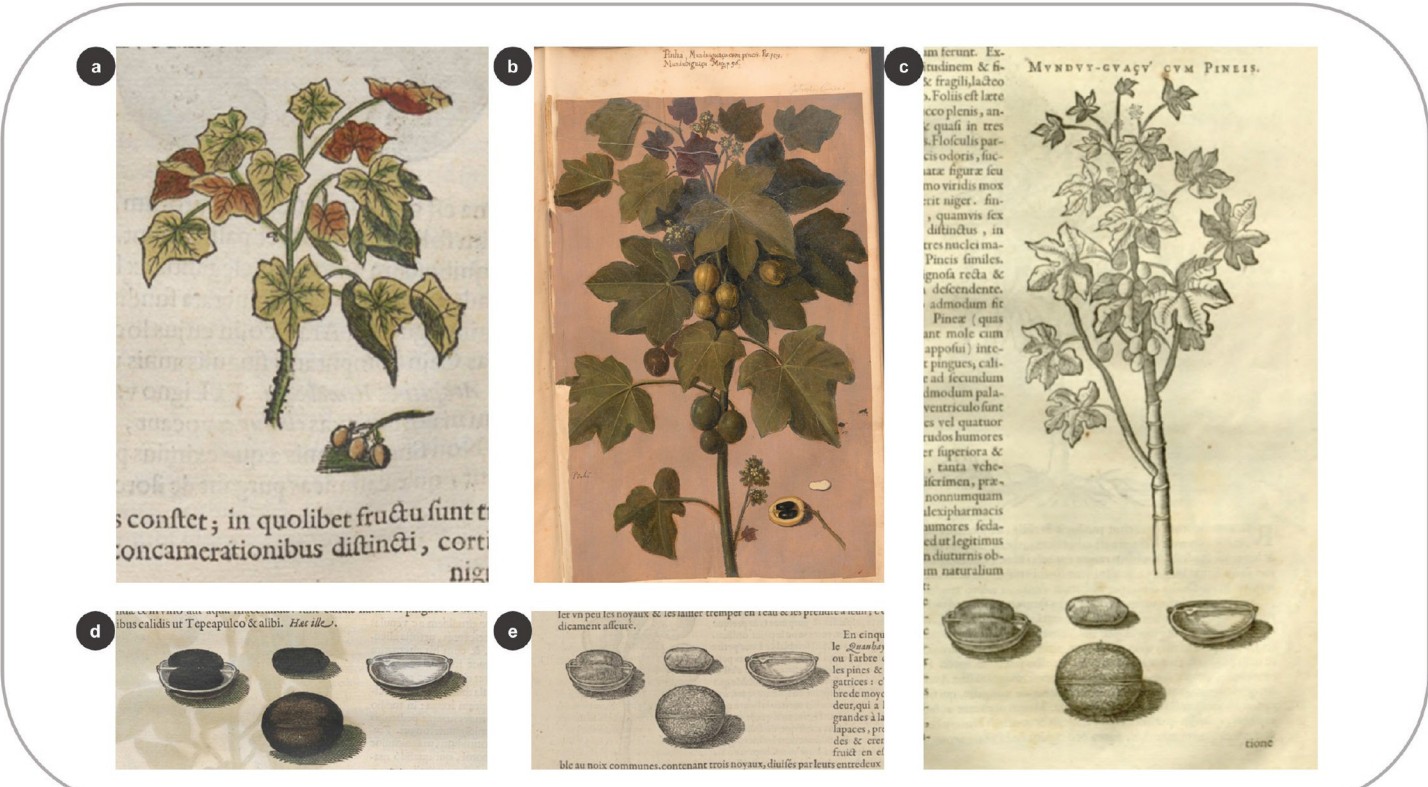

**Fig 10. Combining multiple sources to create the woodcuts for the natural history books.** (a) Woodcut of *Jatropha curcas* L. in the HNB (Marcgrave 1648: 96) (b) Oil painting of the same species in the *Theatrum* (f. 199) (c) Woodcut of this plant and its seeds in Piso (1658: 179) (d) Woodcut of *J. curcas* seeds in the HNB (Marcgrave 1648: 97) (e) Original woodcut of the seeds (De Laet 1640: 136).

lateral branch (Fig 11A). In contrast, the *Theatrum* illustration lacks flowers but displays two open fruits full of red seeds (Fig 11B). These seeds were one of the earliest trade goods exchanged between Indigenous peoples and Europeans in South America and were exported to Europe in the mid-sixteenth century and used as a dye, colorant, and cosmetic [55, 56]. The two images differ significantly, and although the fruits of *B. orellana* were of great economic relevance, De Laet included the woodcut with the less showy and accurate image of them, possibly made after a drawing by Marcgrave. In the IURNM, Piso combined the HNB woodcut with a new woodcut of *B. orellana* fruits (Fig 11C). To make this add-on, he copied the image of a fruiting branch from *Exoticorum Libri Decem* by Clusius (36: 74) (Fig 11D) and added an open fruit with seeds (Fig 11C). Curiously, it seems this fruit was "removed" from Clusius' leftmost part of the branch, opened in half, and laid under the figure (Fig 11C and 11D). Clusius obtained the original branch from the aristocrat and *naturalia* collector Pieter Garet (c.1552/5-1631), who wrote to the botanist that Brazilian Indigenous peoples used the seeds to color their bodies red [57].

**3.3.2 Combining different stages of plant life.** Several woodcuts show flowering and fruiting stages depicted together. For example, in the HNB, the woodcut of *Crateva tapia* L. bears a fruiting branch, including an open fruit with seeds (Fig 12A). In the same woodcut, we also see the inflorescence and the tiny (immature) fruits, characteristic of the Capparaceae species (Fig 12A). Similar to what C.F.P. von Martius observed for *Eugenia uniflora*, *C. tapia* in the *Theatrum* is represented by two illustrations glued in the same folio, each depicting a fruiting and a flowering branch (including the fruit buds) which were combined in one plant image in the HNB (Fig 12B).

Flowering and fruiting stages of *C. tapia* can overlap in nature [58] (example in Species Link: HUEFS 134255, identified and collected by Lyra-Lemos R.P. Alagoas, Brasil), but this is not always the case (Fig 12C). However, the oil paintings show the different fertile structures separately, which must belong to branches with different reproductive stages or the same species collected at other times. Either way, the woodcut in the HNB–similar to those two separate illustrations–conveyed more botanical information in one image and saved money during printing. After all, preparing and including images in treatises was an expensive endeavor in the early modern period [59, 60], as it is today.

Several images merged flowering and fruiting stages, both in the *Theatrum* and the HNB, such as *Annona montana* Macfad. and *Byrsonima cydoniifolia* A. Juss (S1 Appendix: 331, 411), of which the reproductive stages can appear simultaneously in nature (https://specieslink.net/search/). Hence, we cannot infer whether these images were based on separate flowering and fruiting individuals–as was the case for *C. tapia*. We noticed these species were drawn with more leaves in the *Theatrum* than in the HNB. In contrast, other images are represented the other way around: *Paullinia pinnata* L. in the *Theatrum* (f. 283), whose infructescence and open fruit resemble the woodcut in the HNB (33: 114), shows many more (and slightly different) leaves (S1 Appendix: 61).

## 3.4 Connections between De Laet's manuscript and the woodcuts

When connecting the images and descriptions in De Laet's manuscript to the corresponding species and woodcuts in the HNB (S3 Data), we find 388 plant descriptions and 165 plant entries with the word *Icon* next to the entry (S3 Data). Out of these entries, 20 have written "in lib. [x number]" (*in lib* meaning "in the book"). The entry on *Mureci* (*Byrsonima cydoniifolia* A.Juss.) lacks the word *Icon*, but it has "in lib. 93" written next to it; thus, it was likely associated with a specific drawing kept in a notebook and numbered 93. Out of these 166 entries (*Icon* ones plus the entry on *Mureci*), we identified 164 species; two remained unidentified as

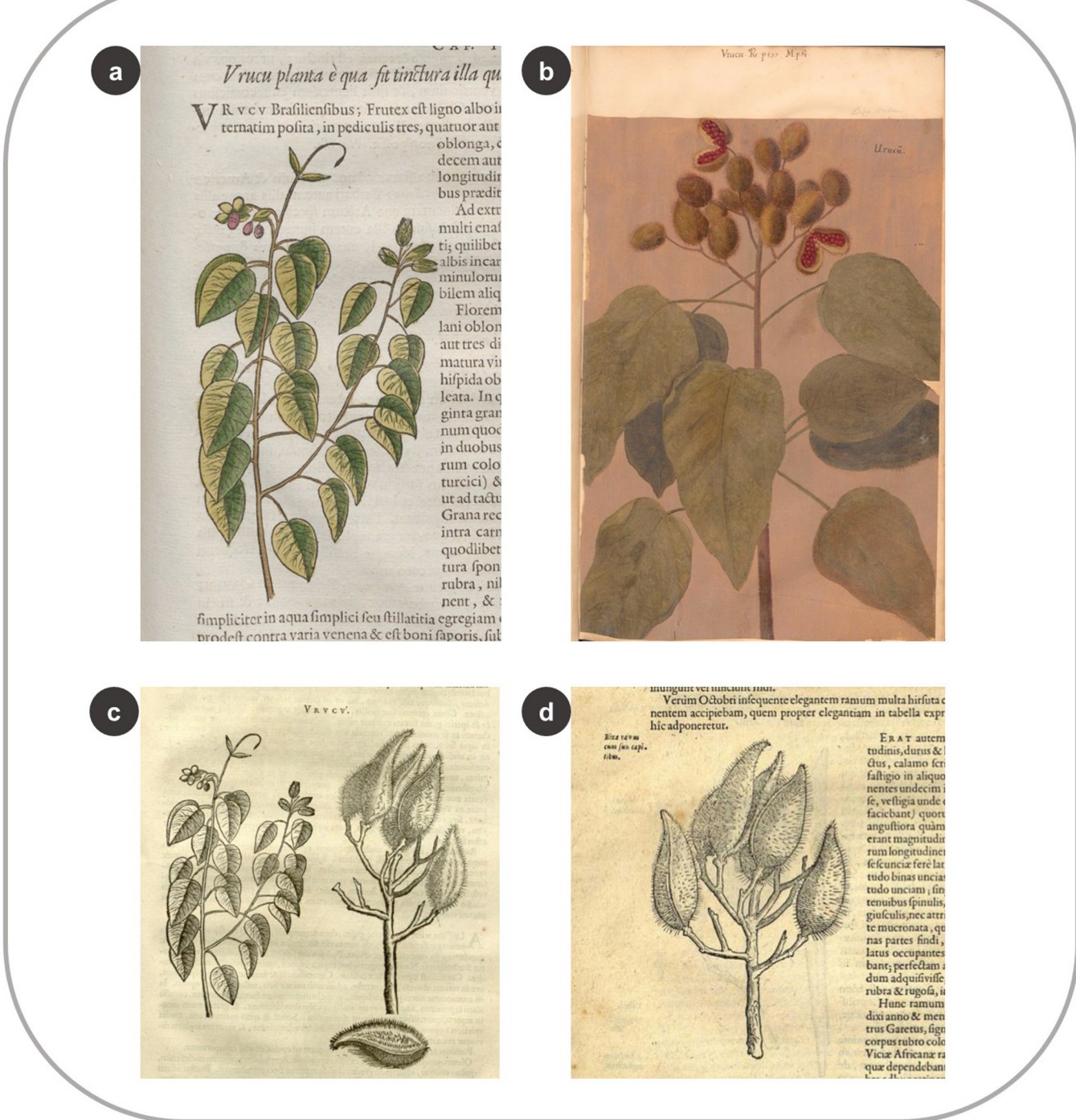

**Fig 11. Using multiple sources to create the plant woodcuts by De Laet and Piso.** (a) Woodcut of *Bixa orellana* L. in the HNB (Marcgrave 1648: 61) (b) Oil-based illustration of *B. orellana* in the *Theatrum* (f. 95) (c) Woodcuts of the same species in the IURNM (Piso 1658: 133) (d) The branch of *B. orellana* in Clusius' treatise (1605: 74).

we could not match them to the descriptions in the HNB (S3 Data). There are 149 *Icon* entries (including *Mureci*) whose plant descriptions include a woodcut in the HNB. Surprisingly, we found 15 entries containing the word *Icon*, but the species described in those entries are not depicted by a woodcut in the HNB. The drawings could have existed, but De Laet did not have

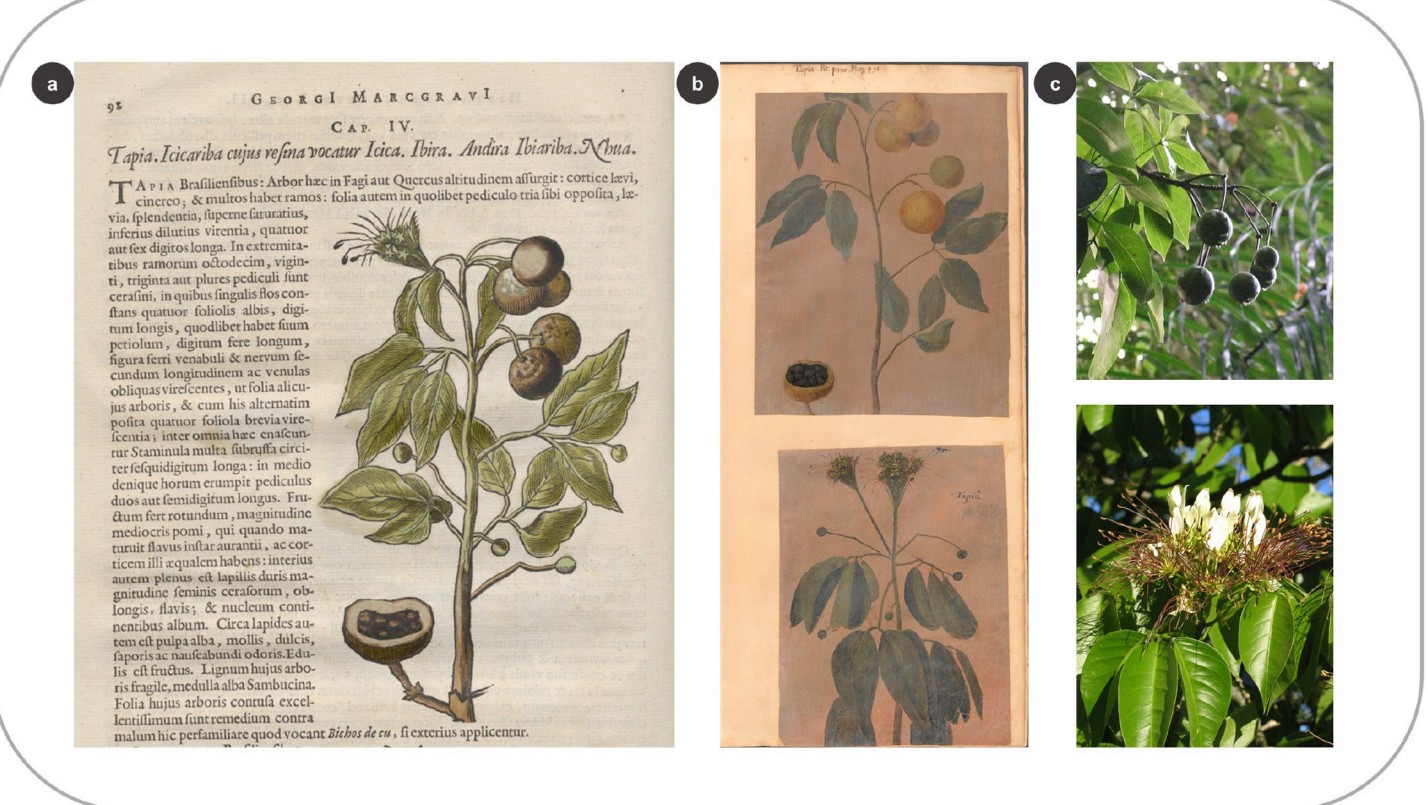

**Fig 12. Combining different stages of plant life in one image.** (a) Woodcut of *Crateva tapia* L. in the HNB (Marcgrave 1648: 98) (b) Fruiting branch of the same species in the *Theatrum* (above) and flowering branch (below) (f. 113) (c) Fruiting branch of *C. tapia* (above), and flowering branch (below) (by T. Leão and A.S. Farias-Castro, in: www.flickr.com).

access to them. For instance, *Tapyracoaynana* (*Cassia grandis* L.f.) in De Laet's manuscript bears the word *Icon in lib. 13* (S3 Data), but this species has no image in the HNB. Instead, De Laet added the vernacular name and description of *C. grandis* next to the description and the woodcut of *Byrsonima sericea* DC., which was made after Marcgrave's herbarium voucher (4: 134). In the IURNM, confused by this arrangement, Piso matched the woodcut of *B. sericea* with the description of *C. grandis* under the name *Tapyracoaynana*, but he also added the woodcut of a pod, presumably belonging to *C. grandis* (42: 158). It is uncertain whether he created this new image after someone else's treatises or after a different species (as he usually did when he did not use the HNB woodblocks) (S2 Appendix: 43–44). Ultimately, he could have had access to the image De Laet referred to in his manuscript.

Noteworthy are the plant entries of this manuscript and their association with the *Theatrum* (S3 Data). Most species (109 spp., 66%) tagged with *Icon* (thus mostly have a woodcut in the HNB) are absent in the *Theatrum*. The remaining (57 spp., 34%) have a correspondent species within the oil paintings, and around half of their woodcuts (27 spp.) are similar to the *Theatrum* (S1 Appendix), with a few of them (8 spp.) with "in lib. [x number]" written (S3 Data). We also found this addition in 13 of the *Icon* entries, but the corresponding woodcuts are not similar to the *Theatrum* images. One of the entries reads *Icon* [. . .] *p. 172* and corresponds to *Furcraea hexapetala* (Jacq.) Urb. In the HNB, De Laet reused the woodcut from his treatise on the Americas (S1 Appendix: 321–322), although number 172 does not match the page number for the same image in the editions we reviewed (26: 666, 27: 608).

By studying how the 26 images in De Laet's manuscript are arranged, we observed that nine drawings in the *verso* folios do not bear the word *Icon* in their descriptions in the *recto*, but they have similar names. Hence, it seems the drawings were glued to the *verso* folios in De Laet's manuscript after the completion (perhaps by another person) by matching the vernacular names written in the recto with the plant descriptions. An exception is *Lecythis pisonis* L., whose lead drawing lacks a name and is placed elsewhere (S3 Data).

Eight of these figures are represented with an identical woodcut in the HNB [5, 9], of which five are only in Piso [32]. This arrangement is strange, as De Laet presumably used this text to elaborate on Marcgrave's chapters on plants. Eight images (four pencil-based and four proof woodcuts) resemble the *Theatrum* illustrations (non-reversed) (S3 Data).

## 3.5 Connections between the images and their vernacular names

All the HNB woodcuts that look similar to their correspondent taxa in the *Theatrum* share the same or similar (cognates) vernacular names (S1 Data). All the entries with the remark *Icon in Lib x* in the manuscript (n = 8) are associated with plant taxa that bear similar images and names in both sources (S3 Data). There are some peculiarities: *Camara uuba* in De Laet's manuscript bears *Icon in lib*. 69 next to it (Fig 13A). Therefore, we would assume that the described plant included a woodcut in the HNB copied from a notebook numbered 69. While the description corresponds to *Calea elongata* Baker [7], the woodcut in the HNB is very similar (in reversed format) to the *Theatrum* illustration for *Lantana camara* L. (Fig 13B). The oil painting bears the name *Camaràuna* (Fig 13C), presumably written by the artist in question (Fig 13D). De Laet warned the reader in the HNB, "the image we give here [with *C. elongata* description], even though we found it under the name *Camara uuba*, seems to be from another *Camara*, which the author [Marcgrave] mentioned before" (4: 6) (Fig 13E).

Most of the taxa with different vernacular names in the HNB and the *Theatrum* are also illustrated differently, except for three species. The woodcuts and illustrations of *Spighelia anthelmia* and *Dorstenia brasiliensis* Lam. look alike but do not bear any name in the oil paintings. These were likely cut-out, as we observed for several oil paintings before being glued into the bound collection (19: 275). *Samanea saman* (Jacq.) Merr. is named *Guaibí pocaca biba* in the HNB (4: 111) and *Nhuatiunana* in the *Theatrum* (f. 399). Since their images are only slightly similar, with similar inflorescences but different leaves and spatial composition, this weakens the possibility that they originated from the same source.

Another species with different local names in the HNB and the *Theatrum* and represented with a different image is *Schinus terebinthifolia* Raddi. A fruiting branch in the HNB (Fig 14A) accompanies the description of this species under the names *Aroeira* and *Lentiscus*. The woodcut, however, belongs to another taxon. De Laet reused the figure (Fig 14B) from his book on the Americas, in which he described a Peruvian tree (27: 327). He copied this image after the treatise *Curae Posteriores* (37: 94), in which Clusius commented on the work of the Spanish physician Monardes on American plants (47: 39). This woodcut represents *Schinus molle* L. (not a Brazilian species) which plant material and knowledge were circulating in Clusius' network of physicians and *naturalia* collectors since the beginning of the seventeenth-century [61].

Ten years later, the woodblock must have been damaged or disappeared, as Piso [3] used a slightly different image based again on De Laet's and Clusius' woodcuts (Fig 14B and 14C), instead of the legitimate species, as the one present in the *Theatrum* (Fig 14D). This oil painting includes the name *Cambuí* (Fig 14D), which we found in the HNB, but for a different species: *Eugenia involucrata* DC. (32: 82). Whoever wrote the reference above the illustration of *S. terebinthifolia* in the *Theatrum* pointed out the woodcut of *E. involucrata*. However, they noticed the discrepancy between the image and the description (Fig 14D). The names *Cambuí* and

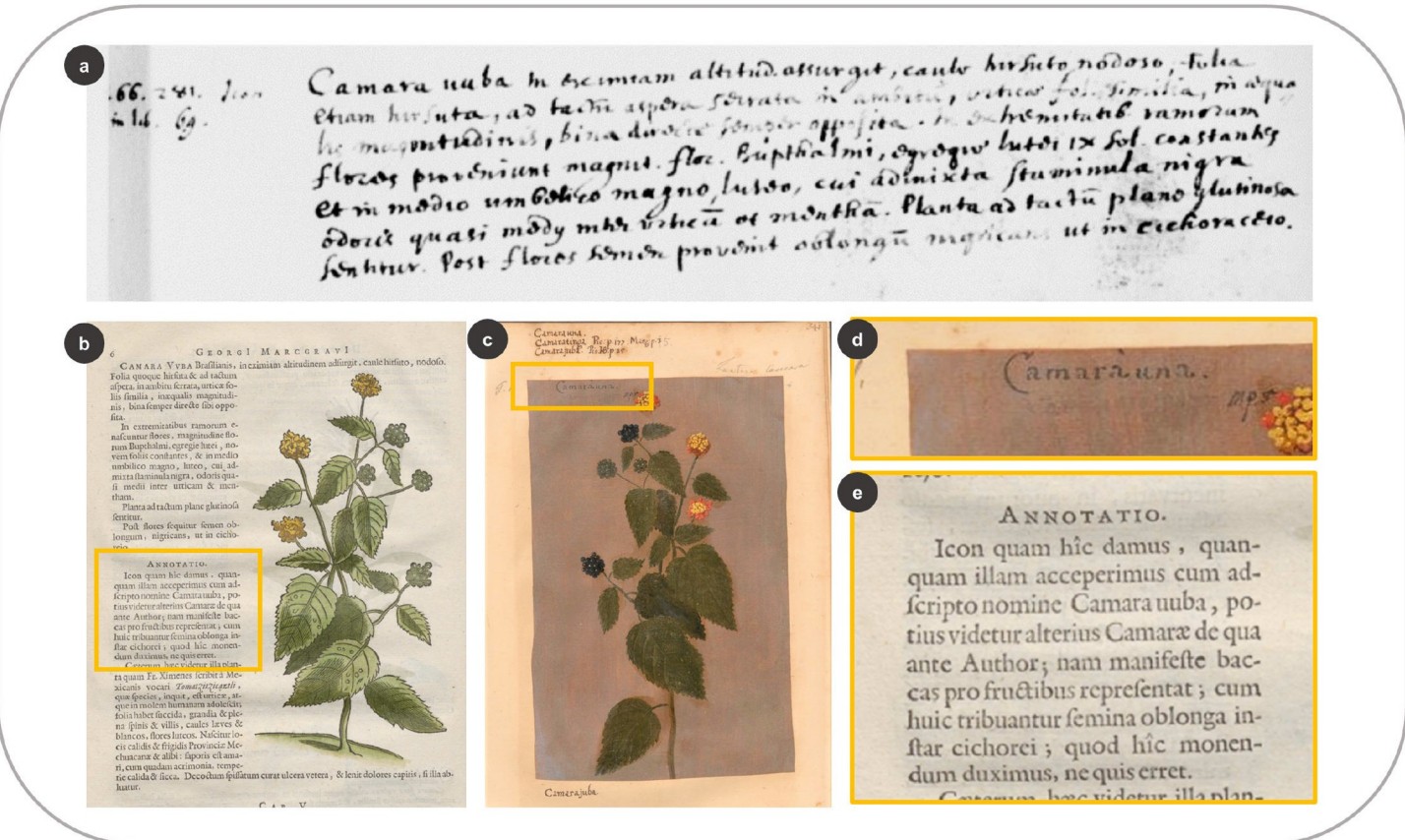

**Fig 13. Linking the woodcuts in the HNB with De Laet's manuscript and the *Theatrum*.** (a) Description of *Calea elongata* Baker next to *Icon in lib. 69* in De Laet's manuscript (f. 5r) (b) Woodcut of *Lantana camara* L. in the HNB (Marcgrave 1648: 6) (c) Oil painting of *L. camara* in the *Theatrum* (f. 341) (d) Vernacular name written on the illustration (e) De Laet's annotation on the use of this image (Marcgrave 1648: 6).

*Aroeira* were used for *S. terebinthifolia* in seventeenth-century Dutch Brazil and are used for the same plant today (Dataplamt, accessed 23.09.22). *Cambuí/i* is currently used for several species within the families of Anacardiaceae, Myrtaceae, and Fabaceae (Dataplamt, accessed 23.09.22).

Overall, if the names differ for the same species, most images do not bear strong resemblances. Nevertheless, the same or similar names can be associated with different images for the same taxa, such as *Bixa orellana*, named *Urucu* in both the HNB and the oil painting (Fig 11). Interestingly, most of the vernaculars in the oil paintings are written in a Tupi-based language, and just a few originated from an African language. This trend contrasts with the HNB, which, apart from the Tupi names, often provides Portuguese and, to a lesser extent, African-based, Dutch, Spanish, and Latin plant names [62].

## 4 Discussion

### 4.1 "Call me by your name": Organizing the Brazilian flora by its Tupi names

Arranging the visual and textual material to create these natural history treatises was an arduous task. As the systematic binomial nomenclature had not yet been established, De Laet, Mentzel, and other scholars who used the HNB as a reference for tropical plants, such as C.F. P. von Martius, struggled to organize the megadiverse Brazilian flora by their vernacular

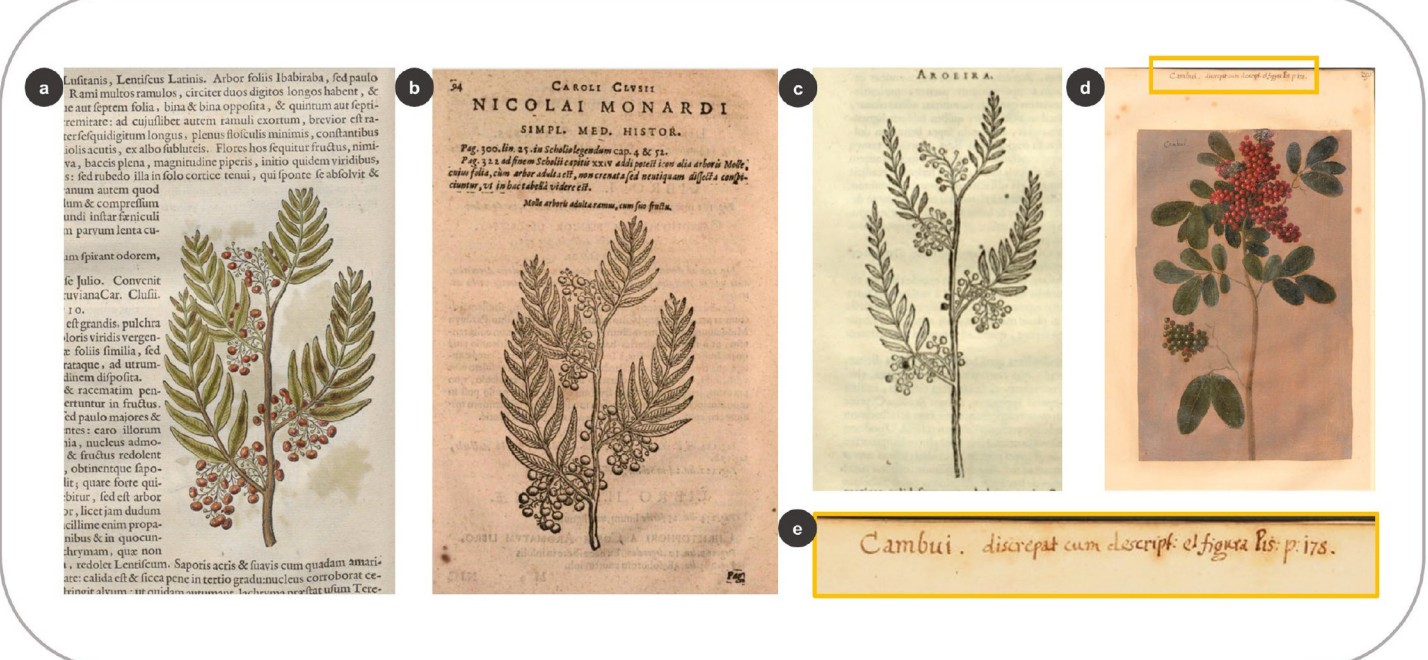

**Fig 14. Different vernacular plant names for the same species resulted in mismatching images and descriptions.** (a) Woodcut of *Schinus molle* L. (*Aroeira*) in the HNB with a description of *S. terebinthifolia* Raddi. (Marcgrave 1648: 90) (b) the same woodcut of *S. molle* in Clusius (1611: 94) (c) a similar woodcut of *S. molle* in Piso (1658: 132) (d) Illustration of *S. terebinthifolia* in the *Theatrum* (f. 295) (e) Close-up of the annotation in the *Theatrum* (f. 295) which includes the name *Cambuí* and the misleading reference to Piso (1658: 178).

names [13]. They cross-referenced the different available sources, just as we did in this study–although we used the scientific names first.

The difficulties of associating the visual material with the right vernaculars and their corresponding descriptions contributed to the development of taxonomy in combination with the flow of exotic plants (either dried, alive, or illustrated) to botanical gardens and private collectors [63, 64]. Linnaeus used the descriptions of Marcgrave and Piso in his *Systema Naturae* [65], and he often used the Tupi Indigenous names as generics or specific epithets [66]–like for *Crateva tapia*, in which *tapia* is a Tupi-based vernacular reported by Marcgrave.

That most of the images are labeled with Tupi-based and a few African plant names imply that artists, naturalists, and Indigenous and enslaved Africans had worked in close connection with each other, as it occurred for the creation of the textual sources in the HNB [31]. Piso stated in his preface that he "submitted to the examination and practice everything that out of the vast theater of nature [he] observed or received by the Indigenous veterans" [3: 6, 41: 8]. Moreover, the painter traveling with Piso into the interior could have been an enslaved African. All members of Johan Maurits's retinue were entitled to personal assistants, as is stated in a document from the WIC at the National Archives in Den Haag [67]. To what extent those contributions were forcibly taken from the local inhabitants is not explicitly mentioned in the archives or natural history books. Nevertheless, such connections were certainly uneven–and likely unjust–due to the structural violence that prevailed in the colony. The military activities of the WIC, which ensured the company and its stakeholder's economic profitability overseas, were sometimes combined with scientific expeditions [1]. Slavery and armed operatives to subjugate local peoples who did not ally with the WIC were part of these activities [68, 69]. It is essential to notice that the Western scholars behind the HNB production oversimplified

Indigenous and African languages under a few categories (such as "Tupi" or "Tupinambá," a language belonging to the administrative and ecclesiastic *lingua franca*). Nevertheless, Indigenous groups spoke more than a thousand languages before the arrival of the Portuguese and their genocide, exposure to foreign diseases, and displacement (https://pib.socioambiental. org/en/Languages). Collaborative linguistic projects with modern Indigenous communities in Brazil, such as those by Da Cruz and Neiva Praça [70], highlighted the connections (i.e., cognates) in plant and animal names between the HNB and three Indigenous communities in Brazil. These studies have proven crucial to increasing access to these historical collections (often only available to the scholarly public) and recognizing the agency of those whose heritage may be connected to them.

## 4.2 Visual knowledge-making processes in the HNB and the IURNM

By analyzing the plant woodcuts, we observed that various models were used to represent the different plant species: freshly picked plants, living individuals, dried fruits, seeds, branches, or herbs, sometimes preserved as herbarium vouchers. Not all images originated in the surroundings of Johan Maurits' palace [22], as plants were gathered in various places and by different people, on several occasions by the Indigenous Brazilians. To portray all fertile stages of a plant was important. Still, sometimes, this aim failed due to inconveniences during the expeditions or the journey back to Europe or because seeds failed to germinate or grow into plants in Holland. Exsiccates were crucial when producing botanical books, allowing authors to compare the specimens with the engravings [71]. However, only a tiny fraction of the woodcuts in the HNB were based on the vouchers collected by Marcgrave (C). Unlike the Renaissance authors of popular herbals such as Fuchs (1501–1566) or Matthioli (1501-c. 1577), who remained in control of publications and could correct botanical mistakes [72], Marcgrave could not intervene to review his draft chapters or his published work because of his premature death. Instead, De Laet, who had never crossed the Atlantic and did not have botanical training, took the lead in assembling the notes and images given to him by Johan Maurits to create the HNB.

This material originated from diverse people who specialized in different subjects but whose skills were connected. Marcgrave was trained as a botanist and astronomer, but he also made drawings and retrieved medicinal plant knowledge from the native population, just as Piso did. Marcgrave, however, was not only interested in the utilitarian value of the flora. Hence, some of his woodcuts belong to species he did not report as useful in the HNB. These images were not represented in the IURNM (S2 Appendix) because the physician aimed to create a more pragmatic field guide. Piso stated that he made his book "with engravings copied after nature [*iconibus ad vivum depictis*], not only for the delight and admiration of the reader but, above all, to serve the doctors and the sick" (3: 47). This statement is partly true: his images were not original *sensu strictu* because he chiefly reused the same woodblocks used for the HNB. Still, most of them originated from the floristic studies in Brazil by his colleague Marcgrave.

To create the HNB, when plant drawings, sketches, or illustrations from Brazil were available, De Laet commissioned figures carved onto the woodblocks. To organize the correct plant images with their description, he often looked at the vernacular names that accompanied those sources, as we observed for the *Theatrum*, in which the overlapping images with the HNB shared the same plant names. However, several accurate botanical representations in the oil paintings, whose names matched those in the HNB, were not carved onto the woodblocks. Other images, such as drawings from Brazil of lesser artistic quality, must have been already available and ended up becoming the basis for the woodcuts. De Laet would match the plant names documented in the field (as seen in De Laet's manuscript) with other sources (i.e., Marcgrave's herbarium, *Theatrum*, herbal treatises, etc.) only if these images were lacking.

Then, he would order the design of a plant drawing to be transferred into a woodblock. This strategy allowed the editor of the HNB to save time and money as he did not have to create a new figure if these were present. When producing the IURNM, even though Piso reused most of the HNB woodcuts, he also added several modifications, especially for the trees, so he altered the woodblocks or made new ones based on the HNB images.

Following Chen [73], who discussed the role of woodblocks in generating visual language in the early modern period, we argue that the HNB created a new visual language by including many images of plant species never seen before in European scholarly circles. Those images constituted a legitimate repertoire later borrowed by others, such as Worm [54] or Piso [3], whose treatise resulted in an accumulation, rather than an innovation, of visual knowledge by replicating the images published in the HNB a decade before. Piso also copied the woodcuts from other authors, especially those used by the Plantin publisher's house in the sixteenth and seventeenth centuries (see several examples in S2 Appendix), even though these plant images were often based on different locations. He sometimes even used images of Mediterranean, Asian, and African species that did not occur in Brazil then or were not reported in the HNB.

### 4.3 Provenance of sources for the plant woodcuts

Throughout this research, we answered the question Whitehead and Boeseman [5] posed on to what extent the *Libri Picturati* images served as models to elaborate the woodcuts of the HNB. The *Libri Principi*'s plant images were not used as the basis for any woodcut in Marcgrave's and Piso's treatises. This fact contrasts with the animal woodcuts in the HNB, which strongly resemble the watercolors in the *Libri Principis* [20, 24]. In contrast to what is argued in previous research [5, 22, 24], the *Theatrum* had some influence but did not constitute the main basis for the plant woodcuts, as only one-third resemble the oil paintings. Hence, if Eckhout made the oil paintings, as indicated by Brienen [22], we cannot entirely attribute the agency behind the woodcut's design to the Dutch painter. We found 163 botanical woodcuts in the HNB (from Piso and Marcgrave's chapters on plants) that cannot be traced to the *Libri Picturati*, Marcgrave's herbarium vouchers used by De Laet, or other scholars' works. Were these woodcuts made after Marcgrave's drawings? This number represents nearly half (47%) of the drawings that Marcgrave mentioned he had made in his letter to De Laet in 1640 [28]. Considering that the naturalist kept working in Brazil until 1643, we would expect a larger number of drawings. Up to this date, despite a few sketches in De Laet's manuscript, whose authorship was presumably attributed to Marcgrave [5, 19, 24], no other records of the naturalist's original drawings exist. If Marcgrave made the pencil drawings in De Laet's manuscript, we must consider that four resemble the *Theatrum* illustrations. Were these drawings part of the nucleus of models made before the oil paintings? Even if Eckhout made several of the oil paintings Mentzel included in the *Theatrum*, we cannot exclude Marcgrave as the likely author of the models used for the oil paintings, or at least some of them. Apart from the pencil drawings glued in the *verso*, several entries (27 species) in De Laet's manuscript with the word *Icon* correspond to the oil painting's images. These could refer to previous drawings present in a numbered notebook created in Brazil. Some plant woodcuts show greater details than their corresponding images in the *Theatrum*, as observed between the animal woodcuts and the *Libri Principis* [24]. Others display less crowded images (e.g., by reducing the number of leaves) than overlapping images in the *Theatrum*. The existence of plant studies previous to the oil paintings, which were later used as models to create the *Theatrum* ones [5, 19, 24], is reasonable. The increased value of the Brazilian imagery lies in the fact that many illustrations and drawings were produced *in situ* [22]. Nevertheless, the oil paintings could have been made in the Dutch Republic, as similarly hypothesized by Johann Horkel (1763–1846) for the

watercolors of the *Libri Principis* [18]. After all, Johan Maurits did not give this iconographic material to the Elector of Brandenburg until 1652, eight years after his return to the Low Countries.

### 4.4 Future research and recommendations

Further studies of the fauna woodcuts in the HNB and IURM and their corresponding species in the *Libri Picturati* are crucial to complement existing studies [5, 6, 14–20, 22–25], to compare our research outcomes and clarify the sources used for the woodcuts. Yet, the location of many sources is unknown or even lost. To solve these mysteries, archival research should be conducted alongside the study of herbaria, libraries, and private collections linked to material originating from Dutch Brazil. Digitizing the known sources is essential to facilitate their analysis without touching the fragile material. Publication of the high-resolution images in an online open format would increase its access to a larger academic community. However, this does not guarantee its dissemination to the broader public. It is pertinent to work towards collaborative projects between Indigenous and Afro-Brazilian communities, researchers, and representatives of the Western institutions that hold this biocultural material. Various historical Brazilian materials have recently been used in successful cross-cultural projects that generated valuable outcomes for the Indigenous communities involved [70, 74–76].

## 5 Conclusions

The repertoire of drawings used to elaborate the HNB and IURNM is incomplete. Creating these treatises can be compared to doing a puzzle with several pieces lacking and the impossibility of going back to gather them. Nevertheless, our systematic analysis reveals new insights about the sources of the woodcuts in these books. The images in these natural history books reflect an intentional effort toward portraying as much botanical information as possible. This goal was achieved by using local people's knowledge of the environment to provide the plant material later captured in images and perhaps to assist further with the artistic process. Moreover, the Tupi-based plant names were crucial for arranging textual and visual sources, sometimes confusing for Western scholars. Our database with all species and vernacular names listed in the HNB and the IURNM, Marcgrave's herbarium, and the *Libri Picturati* (S1 and S2 Data and S1 and S2 Appendices), provides an excellent ground to analyze Western and non-Western plant nomenclature systems and their preservation over time, as well as to review the role of Indigenous and African knowledge embedded in these natural history collections.

Overall, the process of visual knowledge-making differs between the two books: the HNB mostly relied on primary visual sources to depict the flora, while the IURNM relied on secondary ones. Most drawings carved onto the woodblocks arrived from Brazil, sometimes combined with existing images of different provenance and a few new ones made in the Dutch Republic. Yet, the human agency behind such sources requires further attention. Archival studies and collaborative projects with Indigenous and Afro-Brazilian communities and researchers, art historians, and artists could shed light on the missing pieces of this conundrum and the multiple (hidden) histories related to these collections. There is certainly far more than botanical imagery behind the nature portrayed in Dutch Brazil, but to be able to see it, different eyes have to look at it.

## Supporting information

**S1 Appendix. Nature portrayed in images in Dutch Brazil–(S1 Appendix) sources of the plant woodcuts in the *Historia Naturalis Brasiliae* (1648) database.** This PDF file includes the repertoire of woodcut images of the plants depicted in the HNB and their corresponding

images retrieved from older or contemporary sources by cross-referencing their scientific names. This appendix allows us to visualize–among botanical illustrations, herbarium vouchers, plant sketches, and other plant materials that crossed the Atlantic–the potential sources that were used to elaborate the woodcuts in the HNB.
(PDF)

**S2 Appendix. Nature portrayed in images in Dutch Brazil–(S2 Appendix) sources of the plant woodcuts in the *India Utriusque re Naturale et Medica* (1658) database.** This PDF file includes the repertoire of distinctive plant woodcuts included in the IURNM (i.e., those that did not appear in its precedent work, the HNB) and their corresponding images retrieved from older or contemporary sources by cross-referencing their scientific names. This appendix allows us to visualize–among botanical illustrations, herbarium vouchers, plant sketches, and other plant materials–the potential sources that were used to elaborate the woodcuts in the IURNM.
(PDF)

**S1 Data. Sources of the plant woodcuts in the *Historia Naturalis Brasiliae* (Marcgrave and Piso 1648).** This table provides an overview of the plant taxa that were depicted in the HNB. To avoid repetitions, the woodcuts in common to both Marcgrave and Piso for the same species are only depicted in Marcgrave (1648). We analyzed the image's overlaps between the HNB and older or contemporary sources that included botanical images of the same species. Additionally, we added the vernacular plant names as documented in the HNB and the *Theatrum* illustrations to facilitate our comparative analysis between plant names. All the plant images can be visualized in the appendix (S1 Appendix).
(XLSX)

**S2 Data. Sources of the plant woodcuts in the *India Utriusque re Naturale et Medica* (IURNM, Piso 1658).** This table provides an overview of the plant taxa that Piso (1658) depicted with different woodcuts than in the *Historia Naturalis Brasiliae* (HNB, Marcgrave, and Piso 1648) and the presence of the same species in related sources. The similarities among visual sources allow us to hypothesize about the provenance of the woodcuts. These (dis-)similarities can be visualized in an appendix (S2 Appendix). The analogous plant woodcuts the IURNM share with the HNB are analyzed in the main paper based on dataset (S1 Data), with all the plant images displayed in a PDF file (S1 Appendix).
(XLSX)

**S3 Data. Connections between the plant woodcuts in the HNB and the botanical annotations in De Laet's manuscript.** This table shows the interconnections between the plant species listed with "Icon" in the manuscript and the HNB woodcuts in terms of similarities regarding their images and vernacular names. We also analyzed whether these plants are connected to their corresponding images in other sources from Dutch Brazil (e.g., *Theatrum* and Marcgrave's herbarium). The term "Icon" was added to the recto folios. This refers likely to plants sketched or drawn in Brazil and later referenced into a numbered booklet. The verso folios of the manuscript correspond to plant drawings or proof-woodcuts glued to these folios.
(XLSX)

## Acknowledgments

We greatly thank Jagiellonian Library curator Izabela Korczyńska and the British Library (Manuscript Department) for making this research possible by providing us with the digital images of the *Theatrum* and the scanned copies of De Laet's manuscript. Likewise, we are

grateful to information specialist Godard Tweehuysen for giving us access to the HNB kept in Naturalis Biodiversity Center. Our gratitude goes to the plant taxonomists and ethnobotanists who reviewed the woodcuts and botanical illustrations and advised us with their expertise. Thanks to Abisaí J. García Mendoza, Eduardo Hajdu, Jorinde Nuytinck, Nicole de Voogd, Thomas W. Kuyper, and Viviane S. da Fonseca Kruel. Many thanks to Huib Zuidervaart for sharing his exhaustive research on Marcgrave–or, as he found out, Marggrafe. Lastly, big thanks to our colleagues Carolina Monteiro for meticulously digging into the National Archives in Den Haag and sharing crucial data on slavery in Dutch Brazil, and Csilla Ariese for facilitating the use of the Filemaker software and hence contributing to the completion of the visual appendices.

## Author Contributions

**Conceptualization:** Mireia Alcantara-Rodriguez, Tinde Van Andel, Mariana Françozo.

**Formal analysis:** Mireia Alcantara-Rodriguez.

**Funding acquisition:** Mariana Françozo.

**Investigation:** Mireia Alcantara-Rodriguez.

**Methodology:** Mireia Alcantara-Rodriguez, Tinde Van Andel.

**Resources:** Mariana Françozo.

**Supervision:** Tinde Van Andel, Mariana Françozo.

**Visualization:** Mireia Alcantara-Rodriguez.

**Writing – original draft:** Mireia Alcantara-Rodriguez.

**Writing – review & editing:** Tinde Van Andel, Mariana Françozo.

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
