## [Decision Letter · Decision Letter 0]

8 Jan 2023

PONE-D-22-26869

Nature portrayed in images in Dutch Brazil: Tracing the sources of the plant woodcuts in the Historia Naturalis Brasiliae (1648)

PLOS ONE

Dear Dr. Alcantara Rodriguez,

Thank you for submitting your manuscript to PLOS ONE. After careful consideration, we feel that it has merit but does not fully meet PLOS ONE’s publication criteria as it currently stands. Therefore, we invite you to submit a revised version of the manuscript that addresses the points raised during the review process.

We look forward to receiving your revised manuscript.

Kind regards,

Narel Y. Paniagua-Zambrana, M.D.

Academic Editor

PLOS ONE

Journal Requirements:

Additional Editor Comments (if provided):

Dear authors,

We appreciate your patience in waiting, we have been having trouble finding reviewers for your manuscript. The review has ended with the decision to suggest a Minor Review. We thank you for taking into account the comments and suggestions made by the reviewers (attached).

We are waiting for the new version of the manuscript.

Best regards

Reviewers' comments:

Reviewer's Responses to Questions

**Comments to the Author**

1. Is the manuscript technically sound, and do the data support the conclusions?

Reviewer #1: Yes

Reviewer #2: Yes

2. Has the statistical analysis been performed appropriately and rigorously? 

Reviewer #1: N/A

Reviewer #2: N/A

3. Have the authors made all data underlying the findings in their manuscript fully available?

Reviewer #1: Yes

Reviewer #2: Yes

4. Is the manuscript presented in an intelligible fashion and written in standard English?

Reviewer #1: Yes

Reviewer #2: Yes

5. Review Comments to the Author

Reviewer #1: This is a well-written paper and a carefully conducted study, for which I congratulate the authors. I have only a few minor comments. When the authors state that "...played a significant role in the transmission of

(botanical) knowledge" it might be good to clarify that this means "scientific (botanical) knowledge" and not "local knowledge" or LEK.

In the abstract, please make it clear who did the IDs in "we used the scientific identifications of the portrayed plants..." Also, further in the abstract, I do not understand the sentence "availability, economy, and the

Indigenous Tupi-based plant names that accompanied the images were crucial when

arranging the sources". What are "availability" and "economy"? Please be more specific.

Clarify briefly what repository DANS Easy is.

Line 158: specify what "this material" is

M&M: why were categories 2 and 3 not assimilated into one "moderate" category? (from "2) moderately similar (they bear a great resemblance but do not share the same features), 3) slightly similar...). How did these 2 categories influence the data?

Does there exist any info that could enrich this manuscript on why these specific species were selected to make woodcuts?

Reviewer #2: I congratulate the authors for the manuscript. This manuscript is one of the products of a very interesting research project that transcending the boundaries of the natural and human sciences. This is yet another set of data, found about Brazil over 400 years ago, from a growing research area: the analysis of historical documentation integrated into botanical collections in Europe. The data on the analysis of woodcuts under study are original! The article is coherently presented and well written. There is much to commend the manuscript!

Another relevant and original fact is the use of iconographic sources as the basis of the study. These sources are little investigated from an ethnobiological perspective, as in historical ethnobotanical studies. Iconographic sources can bring relevant information about the geographical areas of study, diversity of knowledge that can help a better understanding of the past, that is, the past relationships between people and plants, making it possible to contribute to the understanding of the current state and possibly the future of these relationships (kept? lost? changed?).

• In lines 34 and 35, the authors already demonstrate the relevance of the data set studied, with the study of more than 500 images of plant woodcuts related to more than 420 taxa.

However, I come to bring some suggestions and reflections below:

- Regarding the title 'Dutch Brazil', although a common term in literature, I suggest a reflection! Could you use it as a 'period of Dutch domination in Brazil'?!

'Dutch Brazil' is written both in the title and in lines 54, 56, 59, 69, 97, 134, 205, 579, 720, 750.

Still regarding the usual terms, instead of using 'Indigenous Brazilians' both in the keyword and in the textual part, what do you think about using 'Brazil's indigenous people' or 'indigenous peoples of Brazil'?

- It would be possible to revise the questions or even better to turn them into clearly hypotheses, centered beyond the issue of efforts and the methodological perspective of how to handle and identify the materials (woodcuts and vouchers). Sometimes throughout the manuscript, there is a protocol development bias on how to carry out studies with iconographic sources (woodcuts). I believe there is a wealth of data that could be further explored!

I think I could review and elaborate on these issues further, especially on lines 129 to 138.

The authors could, for example, deepen the central hypothesis about the production of works of illustration for the natural sciences during the period of Dutch domination in Brazil (objects of study, techniques, and materials) considering the importance of Dutch artists and their production techniques of images and objects (over 400 years ago), relating to the European (Dutch) perspectives of the time.

Could the authors deepen even more about the history of art, the history of the Brazilian Northeast, science, ways of life and the perceptions of Europeans.

Seeking to advance and reflect even more on the close relationship between Dutch naturalists, sixteenth-century research and possible “native” Brazilian authors/collaborators, the peculiarities in terms of climate, vegetation, availability of plant resources (shortage of food for long periods, as this region is quite arid and dry).

- Throughout the textual part, it would be interesting to clarify (if possible) a reflection on the possible perspectives of using these findings (of the botanical illustrations) so that they can bring information, as a means of understanding and presenting the indigenous and African heritage in new ways. There is little mention of 'Enslaved Africans', 'Indigenous Brazilians', as seen in lines 42, 43, 97, 287, 297, 587, 607... They could value this information from the beginning of the manuscript!

- The manuscript envisages a multidisciplinary process, combining biology, art and anthropology (?) It emphasizes the methodological issue, that is, how the authors treated and analyzed the datasets of woodcuts, documents and plant vouchers.

However, it would be important to contextualize and deepen the linguistic issue of this region under study, of human groups from the past (16th century) and the current ones. The authors could value even more the vernacular names, the indigenous names of the plants and possible associated data.

- Another interesting aspect would be to highlight the importance of these findings for Brazil, mainly for contributing to the history of biodiversity in the Brazilian Northeast.

I also suggest a better characterization of the Brazilian northeast region, this part is quite superficial, mentioned only in line 56. It would be essential to contextualize the region under study, highlighting its climatic peculiarities, vegetation, human groups, ways of life (a summary of the human population, indigenous languages and origin) both from the period of Dutch rule in Brazil and today.

It would be worth characterizing and highlighting the places where the Dutch concentrated their efforts and studies, where they collected and described plant resources (more in Bahia? Pernambuco? coastal vegetation? caatinga? Atlantic forest?).

The manuscript lightly describes the “Northeast” of Brazil, it is interesting to highlight the diversity, abundance and richness of the local flora and culture. The Northeast is rich and diverse, as it encompasses different biomes and plant formations (caatinga, Atlantic forest and coastal vegetation), and of course, this has implications for the availability of species by environment. I suggest better characterizing the region of origin of the woodcuts.

Will this information be relevant to advancing hypotheses related to the versatility of uses, for exemple. Introduced and/or native species in Brazil would be important initially for their food potential and secondly for their medicinal potential)? The northeastern region of Brazil is characterized by long periods of food scarcity (an extremely dry and arid region), could it be that this characteristic provided some bias of interest in certain plants from the 16th century?

- This data crossing information (past and current) is particularly important to assist in future studies of mapping and availability of plant resources, ways of life. I believe that few Brazilians know and/or have access to the history of plant and cultural diversity in Brazil (in the Northeast region) in the 16th century. This information needs wide visibility and appreciation!

- The authors could expand the scope and bring some future of returning the scientific knowledge analyzed here to the indigenous and/or traditional communities of the Brazilian Northeast, especially to young people? Or bring a perspective on how these data can be used to expand knowledge of plant and cultural diversity in Brazil. Or how to promote greater visibility and exchange of information from this data set, how to return this knowledge about these collections (forgotten in European libraries) to Brazilian institutions, indigenous peoples and/or places in the Brazilian Northeast from where they were originally taken? Is there a future perspective that museums/institutions can develop new ways of presenting historical data about plants and cultural diversity?

* I know it's out of this scope, but it's worth thinking about!

- In the methods, it is important to explain more about the historical sources, especially in relation to the transcripts and translations used!

Does it highlight whether original (primary), written or iconographic sources were used to analyze the datasets?

It would be worth reviewing the methodology described for crossing information, especially between lines 158 to 168.

- Would it be possible to list the ethnicities in the database (described in lines 169 to 171) or better highlight in which area of the Northeast and possible ethnicities would be related to certain information (vernacular names)?

Still in methods, in lines 173 to 175, I could better distinguish and explain the difference between 2) moderately similar and 3) slightly similar. It got a little confusing!

As for results, it would be interesting to explore the data further, as initially described here in the review!

For example, in line 206, the finding on the woodcut of Spighelia anthelmia L. was described, and then the data were little discussed in line 545. This species is widely known and used in traditional medicine (the 'worm herb'), treats it if it is a spontaneous species, annual and occurring in the tropical region, it is a potent vermifuge, but there is toxicity ... finally, explore even more the information on the species. Another very important species on the Brazilian coast and known to be medicinal is Schinus terebinthifolia Raddi (but there are doubts about the species of Schinus in the northeast, there are possibly two species of mastic – Schinus, S. terebinthifolia and S. molle). It would be worth a deepening on the highlighted species.

6. PLOS authors have the option to publish the peer review history of their article (what does this mean?). If published, this will include your full peer review and any attached files.

Reviewer #1: No

Reviewer #2: No

---

## [Author Response · Author response to Decision Letter 0]

13 Apr 2023

Response to Reviewers [PONE-D-22-26869]

Dear Dr. Narel Y. Paniagua-Zambrana,

Many thanks for allowing us to revise our manuscript entitled: "Nature portrayed in images in Dutch Brazil: Tracing the sources of the plant woodcuts in the Historia Naturalis Brasiliae (1648)", PONE-D-22-26869. 

Below, we respond to each point raised by the reviewers. The numbered lines correspond to the manuscript with track changes. 

Reviewers' comments:

1. Is the manuscript technically sound, and do the data support the conclusions?

Reviewer #1: Yes

Reviewer #2: Yes

2. Has the statistical analysis been performed appropriately and rigorously?

Reviewer #1: N/A

Reviewer #2: N/A

3. Have the authors made all data underlying the findings in their manuscript fully available?

The PLOS Data policy requires authors to make all data underlying the findings described in their manuscript fully available without restriction, with rare exceptions (please refer to the Data Availability Statement in the manuscript PDF file). The data should be provided as part of the manuscript or its supporting information, or deposited to a public repository. For example, in addition to summary statistics, the data points behind means, medians and variance measures should be available. If there are restrictions on publicly sharing data—e.g. participant privacy or use of data from a third party—those must be specified.

Reviewer #1: Yes

Reviewer #2: Yes

4. Is the manuscript presented in an intelligible fashion and written in standard English?

Reviewer #1: Yes

Reviewer #2: Yes

5. Review Comments to the Author

Reviewer #1: This is a well-written paper and a carefully conducted study, for which I congratulate the authors. I have only a few minor comments. When the authors state that "...played a significant role in the transmission of (botanical) knowledge" it might be good to clarify that this means "scientific (botanical) knowledge" and not "local knowledge" or LEK.

Response: We thank the reviewer for these positive comments. We have changed the sentence "transmission of (botanical) knowledge" into "transmitting scientific botanical knowledge" in the abstract and line 34.

Reviewer #1: In the abstract, please make it clear who did the IDs in "we used the scientific identifications of the portrayed plants..." 

Response: We have changed this sentence in the abstract (line 33) to: "To do so, we used our previous botanical identifications of the portrayed plants, published sources, and digital archival material."

Reviewer #1: Also, further in the abstract, I do not understand the sentence "availability, economy, and the Indigenous Tupi-based plant names that accompanied the images were crucial when arranging the sources". What are "availability" and "economy"? Please be more specific. 

Response: We change this sentence to specify better what we mean by that. “Substantial differences in the visual-making methodology exist between the two treatises (1648, 1658). In the first book, most of the images were available from Dutch Brazil and carved into the woodcuts, while most of these woodcuts were reused in the second one” (lines 38-40).

Reviewer #1: Clarify briefly what repository DANS Easy is. 

Response: DANS Easy is a virtual repository. This is used to archive research data to make it accessible worldwide to researchers and interested parties (see more at https://dans.knaw.nl/en/data-services/easy/).

Reviewer #1: Line 158: specify what "this material" is.

Response: We have now changed this sentence to "To trace the origins of the woodcut images in the HNB and IURNM, we also…" (line 169).

Reviewer #1: M&M: why were categories 2 and 3 not assimilated into one "moderate" category? (from "2) moderately similar (they bear a great resemblance but do not share the same features), 3) slightly similar...). How did these 2 categories influence the data? Does there exist any info that could enrich this manuscript on why these specific species were selected to make woodcuts?

Response: We have elaborated better these distinctions in the text ( Lines 191-198) “We distinguished four categories based on the level of details the artists copied from the Libri Picturati's images into the woodblock's models: 1) very similar (woodcut and illustration share [almost] the same features), 2) moderately similar (they bear a remarkable resemblance but do not share exactly the same features), 3) slightly similar (they are different in most features yet some details, such as the inflorescence, the fruits, a few leaves, etc., look-alike, and 4) different (not enough similar features between images to assume any correlation). The more similar a woodcut and its corresponding species in the Libri Picturati were, the more probably the woodcut was made after the image, or they both originated from the same source”.

Referring to Albizia saman (an in-text example of a slightly similar woodcut compared to the oil painting in the Theatrum), we added in the results (Lines 609-611), "Since their images are only slightly similar, with similar inflorescences but different leaves and spatial composition, this weakens the possibility that they originated from the same source." 

Those species were selected to make the woodcuts because their names matched the vernacular names that accompanied the descriptions (as seen in De Laet's manuscript). Often, the names in the manuscript and the illustration in the Libri Picturati matched (such as Bixa orelllana – line 645). However, De Laet must have had an already made drawing (likely those made by Marcgrave in the field), and he used those to accompany the description in the HNB. We think that in that way, he saved the costs of commissioning a new drawing out of the oil paintings (or the models before them) (see in the manuscript lines 719-732). 

Reviewer #2: I congratulate the authors for the manuscript. This manuscript is one of the products of a very interesting research project that transcending the boundaries of the natural and human sciences. This is yet another set of data, found about Brazil over 400 years ago, from a growing research area: the analysis of historical documentation integrated into botanical collections in Europe. The data on the analysis of woodcuts under study are original! The article is coherently presented and well written. There is much to commend the manuscript! Another relevant and original fact is the use of iconographic sources as the basis of the study. These sources are little investigated from an ethnobiological perspective, as in historical ethnobotanical studies. Iconographic sources can bring relevant information about the geographical areas of study, diversity of knowledge that can help a better understanding of the past, that is, the past relationships between people and plants, making it possible to contribute to the understanding of the current state and possibly the future of these relationships (kept? lost? changed?). In lines 34 and 35, the authors already demonstrate the relevance of the data set studied, with the study of more than 500 images of plant woodcuts related to more than 420 taxa. 

However, I come to bring some suggestions and reflections below:

Response: We are grateful to the reviewer for these positive remarks.

Reviewer #2: Regarding the title 'Dutch Brazil', although a common term in literature, I suggest a reflection! Could you use it as a 'period of Dutch domination in Brazil'?! 'Dutch Brazil' is written both in the title and in lines 54, 56, 59, 69, 97, 134, 205, 579, 720, 750.

Response: We choose to use the term “Dutch Brazil” because, indeed, as pointed out by reviewer 2, this is the common term used in the literature and therefore it makes our paper more easily accessible/findable for researchers. In the first paragraph of the introduction (line 59), we do explain that this term refers to the Dutch occupation and colonization of part of the Brazilian territory in the seventeenth century. 

Reviewer #2: Still regarding the usual terms, instead of using 'Indigenous Brazilians' both in the keyword and in the textual part, what do you think about using 'Brazil's indigenous people' or 'indigenous peoples of Brazil'?

Response: Regarding the terms Brazil's indigenous people or indigenous peoples of Brazil, we prefer not to use either one of these expressions because both imply possession or belonging ('s and of); that is, they suggest that the indigenous peoples belong to or in Brazil. However, from an archaeological, historical, and anthropological perspective, this is incorrect. Indigenous peoples existed long before the colonization and long before the creation of the nation-state Brazil. For many of them still today, the national borders are less significant than for non-Indigenous peoples, as Indigenous communities transcend borders and have relationships and ways of life that go beyond them. The term Indigenous Brazilians, on the other hand, acknowledges that they were born in or live in Brazil but that they are nonetheless Indigenous and not 'property' of the country. 

Reviewer #2: It would be possible to revise the questions or even better to turn them into clearly hypotheses, centered beyond the issue of efforts and the methodological perspective of how to handle and identify the materials (woodcuts and vouchers). Sometimes throughout the manuscript, there is a protocol development bias on how to carry out studies with iconographic sources (woodcuts). I believe there is a wealth of data that could be further explored!

Response: We have explained in detail how we identified the woodcuts, herbarium vouchers, and botanical illustrations in the Libri Picturati in our previous paper (Alcantara-Rodriguez et al. 2021. Looking into the flora of Dutch Brazil: Botanical identifications of seventeenth century plant illustrations in the Libri Picturati. Scientific Reports, 11(1), 19736.). Since the current paper is already quite long, we decided to cite our previous work on how we did the identifications instead of adding these methodological details again in the current manuscript.

Reviewer #2: I think I could review and elaborate on these issues further, especially on lines 129 to 138.

The authors could, for example, deepen the central hypothesis about the production of works of illustration for the natural sciences during the period of Dutch domination in Brazil (objects of study, techniques, and materials) considering the importance of Dutch artists and their production techniques of images and objects (over 400 years ago), relating to the European (Dutch) perspectives of the time.

Response: Our main goal in this manuscript was to trace the (diversity of) visual sources of the woodcuts in the published treatises HNB and IURNM. The importance of Dutch artists on the European worldview in the seventeenth century is substantial but lies beyond the scope of our research. We cite other authors (Whitehead & Boeseman, 1989; Joppien, 1979; Brienen, 2006) who cover this subject in more detail.

Reviewer #2: Could the authors deepen even more about the history of art, the history of the Brazilian northeast, science, ways of life and the perceptions of Europeans. Seeking to advance and reflect even more on the close relationship between Dutch naturalists, sixteenth-century research and possible "native" Brazilian authors/collaborators, the peculiarities in terms of climate, vegetation, availability of plant resources (shortage of food for long periods, as this region is quite arid and dry).

Response: Although all very interesting subjects, they fall beyond the scope of the current paper. We have published on African, native Brazilian and European food crops in seventeenth century in our previous paper Alcantara-Rodriguez et al. 2019. Plant knowledge in the Historia Naturalis Brasiliae (1648): Retentions of seventeenth-century plant use in Brazil. Economic Botany, 73, 390-404. 

Reviewer #2: Throughout the textual part, it would be interesting to clarify (if possible) a reflection on the possible perspectives of using these findings (of the botanical illustrations) so that they can bring information, as a means of understanding and presenting the indigenous and African heritage in new ways. There is little mention of 'Enslaved Africans', 'Indigenous Brazilians', as seen in lines 42, 43, 97, 287, 297, 587, 607... They could value this information from the beginning of the manuscript!

Response: To acknowledge the presence of the diverse local groups during the activities of Maurits’ in the colony and their asymmetrical power relationships in the seized land, we added at the beginning of the introduction (lines 64-69), “With the support of the WIC, the count commanded a military guard to accompany Marcgrave when collecting fauna and flora (1). At the same time, local people (often Indigenous Brazilians, but also enslaved Africans and Portuguese) brought the specimens to him while he resided in Maurits’ court. The same applied to Piso, who, like the other members of Johan Maurits’ entourage, was entitled to an assistant for himself.”

The reviewer raised a valid point about decolonizing historical texts and narratives. While we acknowledged the unequal power dynamics between European colonizers and Indigenous and enslaved Africans coexisting in the colony, this information is not discussed in-depth in our manuscript. On the one hand, our aim was to trace the origins of the woodcuts. Information on the use of plants by Indigenous Brazilians and enslaved Africans in seventeenth century Brazil was published in our previous paper: Alcantara-Rodriguez et al. 2019. Retentions of seventeenth-century plant use in Brazil. Econ. Bot. 

On the other hand, the making of these texts and paintings is attributed to Western scholars and artists (white, European, middle-upper class men), who created them from their Eurocentric perspective; hence, their narratives prevail in the archival visual and textual material. Now, the knowledge embedded in these sources did come from a more heterogenous context, as indicated in our previous paper. The various Indigenous peoples and enslaved Africans played a role in the knowledge-making processes, and using these illustrations could bring up many plant stories that acknowledge their heritage. Future collaborations with Indigenous and Afro-Brazilians will shed more light on the representation of Indigenous and African heritage at present by looking at historical sources together.

We want to encourage further critical studies that go more in-depth with the issues brought up by the reviewer, as we indicated in lines 682-686: “Our database with all species and vernacular names listed in the HNB and the IURNM, Marcgrave’s herbarium, and the Libri Picturati (S1, S2, S3, S4), provides a good ground to analyze Western and non-Western plant nomenclature systems and their preservation over time, as well as to review the role of Indigenous and African knowledge embedded in these natural history collections”.

Reviewer #2: The manuscript envisages a multidisciplinary process, combining biology, art and anthropology (?) It emphasizes the methodological issue, that is, how the authors treated and analyzed the datasets of woodcuts, documents and plant vouchers. However, it would be important to contextualize and deepen the linguistic issue of this region under study, of human groups from the past (16th century) and the current ones. The authors could value even more the vernacular names, the indigenous names of the plants and possible associated data.

Response: We agree with the reviewer that there is much valuable linguistic data (in the form of local plant names) included in these published and unpublished botanical illustrations from seventeenth century Dutch Brazil. It requires extensive study by linguists to trace these Tupi-related plant names to currently used indigenous plant names in Brazil. In a previous paper, we covered a small part of this interesting field: Geertsma, I. P., Françozo, M., van Andel, T., & Rodríguez, M. A. (2021). What's in a name? Revisiting medicinal and religious plants at an Amazonian market. Journal of Ethnobiology and Ethnomedicine, 17(1), 1-15. 

Recently, linguistic elements of the HNB were discussed with indigenous people who speak Tupi-related languages in Brazil, and the results were published some weeks ago by Aline da Cruz and Walkíria Neiva Praça (2023) Reconnecting Knowledges: Historia Naturalis Brasiliae back to Indigenous Societies. https://www.taylorfrancis.com/chapters/oa-edit/10.4324/9781003362920-8/reconnecting-knowledges-aline-da-cruz-walk%C3%ADria-neiva-praça?context=ubx&refId=57d71bc0-c940-4406-8684-d102d6df66f2

Reviewer #2: Another interesting aspect would be to highlight the importance of these findings for Brazil, mainly for contributing to the history of biodiversity in the Brazilian northeast. I also suggest a better characterization of the Brazilian northeast region, this part is quite superficial, mentioned only in line 56. It would be essential to contextualize the region under study, highlighting its climatic peculiarities, vegetation, human groups, ways of life (a summary of the human population, indigenous languages and origin) both from the period of Dutch rule in Brazil and today.

It would be worth characterizing and highlighting the places where the Dutch concentrated their efforts and studies, where they collected and described plant resources (more in Bahia? Pernambuco? coastal vegetation? caatinga? Atlantic forest?).

The manuscript lightly describes the "Northeast" of Brazil, it is interesting to highlight the diversity, abundance and richness of the local flora and culture. The northeast is rich and diverse, as it encompasses different biomes and plant formations (caatinga, Atlantic forest and coastal vegetation), and of course, this has implications for the availability of species by environment. I suggest better characterizing the region of origin of the woodcuts.

Response: In the present paper, we shortly mention "the sertão, the dry hinterland of northeastern Brazil" in line 350. The natural vegetation, climate, and native inhabitants of the area colonized by the Dutch do not fall within the scope of our current paper. However, we have incorporated these aspects extensively in our previous paper (Alcantara-Rodriguez et al. 2021. Looking into the flora of Dutch Brazil), in which we describe the vegetation types where the plants grew depicted in both the Libri Picturati and the woodcuts and whether they were endemic to Brazil or specific regions of the country.

Reviewer #2: Will this information be relevant to advancing hypotheses related to the versatility of uses, for example. Introduced and/or native species in Brazil would be important initially for their food potential and secondly for their medicinal potential)? The Northeastern region of Brazil is characterized by long periods of food scarcity (an extremely dry and arid region), could it be that this characteristic provided some bias of interest in certain plants from the 16th century?

Response: In our previous paper (Alcantara-Rodriguez et al. 2021. Looking into the flora of Dutch Brazil), we specifically mention plants typical to northeast Brazil that can withstand drought and are used for emergency food by the people living there (like Spondias tuberosa). Comparisons with seventeenth century and current plant use, based on these plant illustrations have been published in our paper in Economic Botany (Alcantara-Rodriguez et al. 2019). 

Reviewer #2: This data crossing information (past and current) is particularly important to assist in future studies of mapping and availability of plant resources, ways of life. I believe that few Brazilians know and/or have access to the history of plant and cultural diversity in Brazil (in the Northeast region) in the 16th century. This information needs wide visibility and appreciation!

Response: We fully agree with the reviewer and have published this already (see our previous remark). 

Reviewer #2: The authors could expand the scope and bring some future of returning the scientific knowledge analyzed here to the indigenous and/or traditional communities of the Brazilian northeast, especially to young people? Or bring a perspective on how these data can be used to expand knowledge of plant and cultural diversity in Brazil. Or how to promote greater visibility and exchange of information from this data set, how to return this knowledge about these collections (forgotten in European libraries) to Brazilian institutions, indigenous peoples and/or places in the Brazilian northeast from where they were originally taken? Is there a future perspective that museums/institutions can develop new ways of presenting historical data about plants and cultural diversity?

* I know it's out of this scope, but it's worth thinking about!

Response: We have also thought about this a lot. Due to covid-19, we could not travel to Brazil during the period in which we did our research (2020-2021). However, our supporting information contains all the visual information on which we based our research, so that will become available to the world with the publication of this paper. Many of these images have never been published before with their correct botanical names.

Reviewer #2: In the methods, it is important to explain more about the historical sources, especially in relation to the transcripts and translations used! Does it highlight whether original (primary), written or iconographic sources were used to analyze the datasets? It would be worth reviewing the methodology described for crossing information, especially between lines 158 to 168.]

Response: In our methods, we mention that we worked with digital images of the visual sources. We visited the original Marcgrave’s herbarium on an earlier occasion (2018) and consulted original volumes of the HNB and IURNM in the Naturalis Library. For the current paper, however, we worked with digital images.

Reviewer #2: Would it be possible to list the ethnicities in the database (described in lines 169 to 171) or better highlight in which area of the northeast and possible ethnicities would be related to certain information (vernacular names)?

Response: We do not know precisely where Marcgrave (and Piso) collected their specimens, as they do not mention this in the written sources for the majority of the species. Hence, we cannot plot these on a map. The few locations indicated in the HNB for specific plants were already highlighted by Almeida (2016). De. Historiae Rervm Naturalivm: ensaios histórico-culturais sobre as ciências biológicas. Recife.

With regard to the Indigenous Brazilians, much has changed since the Dutch left Brazil. People speaking Tupi-related languages around 1640 in Pernambuco have migrated elsewhere, became victims of colonial violence, infectious diseases, or merged with other ethnicities. Therefore, it is not possible to locate peoples in Brazil today who are "the descendants" of those from whom Marcgrave documented the names and plan uses in the 1640s. We have elaborated on this issue in our paper Alcantara Rodriguez, M., Pombo Geerstma, I., De Campos Françozo, M., & van Andel, T. R. (2020). Marcgrave and Piso's plants for sale: The presence of plant species and names from the Historia Naturalis Brasiliae (1648) in contemporary Brazilian markets. Journal of Ethnopharmacology, 259, 112911.

Reviewer #2: Still in methods, in lines 173 to 175, I could better distinguish and explain the difference between 2) moderately similar and 3) slightly similar. It got a little confusing!

As for results, it would be interesting to explore the data further, as initially described here in the review!

Response: We understand this distinction seems a bit confusing. By looking at the illustrations compared to the woodcuts in the digital files we created (S2 and S4), the reader can better visualize the differences between categories. To make it more evident in the text, we added (lines 191-198), " We distinguished four categories: 1) very similar (woodcut and illustration share [almost] the same features), 2) moderately similar (they bear a remarkable resemblance but do not share exactly the same features), 3) slightly similar (they are different in most features yet some details, such as the inflorescence, the fruits, a few leaves, etc., look-alike), and 4) different (not enough similar features between images to assume any correlation). The more similar a woodcut and its corresponding species in the Libri Picturati were, the more probably the woodcut was made after the image, or they both originated from the same source".

In addition, referring to Albizia saman (an in-text example of a slightly similar woodcut compared to the oil painting in the Theatrum), we added in the results (lines 609-611), "Since their images are only slightly similar, with similar inflorescences but different leaves and spatial composition, this weakens the possibility that they originated from the same source." 

The slightly similar images between the HNB and the Libri Picturati only accounted for 9%. After all, there are more differences between corresponding images between the woodcuts and the oil paintings (for the same species) than similarities. Hence, these illustrations later included in the Libri Picturati (or the sketches or drawings made before them) were not the primary sources used to elaborate the woodcuts in the HNB and IURNM (as suggested before). Instead, several drawings from the field in Brazil (likely made by the naturalist Marcgrave) were used as models by De Laet and ten years later reused by Piso. The nuances in the similarity between images do not change our conclusions. However, we thought it was important to make those distinctions based on how much in detail the artists went into when copying the figures that would end up being carved onto the woodblocks.

Reviewer #2: For example, in line 206, the finding on the woodcut of Spighelia anthelmia L. was described, and then the data were little discussed in line 545. This species is widely known and used in traditional medicine (the 'worm herb'), treats it if it is a spontaneous species, annual and occurring in the tropical region, it is a potent vermifuge, but there is toxicity ... finally, explore even more the information on the species. 

Response: Although plant uses of these plants were extensively described in our other papers, we have added a short sentence (line 227), "The common medicinal plant against intestinal worms, Spigelia anthelmia L." Just like elsewhere in the world where this weed grows, it is used against worms, and pretty poisonous as well.

Reviewer #2: Another very important species on the Brazilian coast and known to be medicinal is Schinus terebinthifolia Raddi (but there are doubts about the species of Schinus in the northeast, there are possibly two species of masYc – Schinus, S. terebinthifolia and S. molle). It would be worth a deepening on the highlighted species.

Response: Indeed, Schinus terebinthifolia, known as Brazilian pepper tree or Aroeira vermelha, grows along the Brazilian coast and is the tree with the round leaflets depicted in the oil painting (Figure 13d). S. molle also occurs in Brazil, but more in the south (Rio Grande do Sul) and has very thin leaves (as the woodcut in the HNB has, Figure 13 a). We used the example of S. terebinthifolia to discuss the role of the vernacular names in the making of the HNB. The editor of the HNB, De Laet, mistook it for another species (S. molle), a known medicine elsewhere in South America and depicted in other books on herbal medicine. Surprisingly, the right image of the tree (S. terebinthifolia) was included in the Theatrum (available to him). However, he did not associate it with the description because their names did not match. He had to reuse a previously published image, ultimately using the wrong one, which Piso also mistook ten years later. 

As pointed out by the reviewer, S. terebinthifolia is indeed an important medicinal plant. This fact was highlighted in historical sources, such as the HNB, and its medicinal value has persisted today. In the supplementary electronic material - ESM2 accompanying our previous paper Alcantara-Rodriguez et al. 2019, we indicated the uses of S. terebinthifolia as described in the HNB and the potential correlations with its uses in modern Brazil. The HNB points out that the resin of this tree and the oil squeezed from the fruits act as a poultice to treat cold affections, and the aromatic leaves were used for baths. In modern literature, we found that the Aroeira oil extracted from the bark is used for tumors arising from arthritis or syphilis; the leaves are anti-rheumatic and also used to heal ulcers and wounds.

---

## [Decision Letter · Decision Letter 1]

3 Nov 2023

PONE-D-22-26869R1Nature portrayed in images in Dutch Brazil: Tracing the sources of the plant woodcuts in the Historia Naturalis Brasiliae (1648)PLOS ONE

Dear Dr. Alcantara Rodriguez,

Thank you for submitting your manuscript to PLOS ONE. After careful consideration, we feel that it has merit but does not fully meet PLOS ONE’s publication criteria as it currently stands. Therefore, we invite you to submit a revised version of the manuscript that addresses the points raised during the review process.

We look forward to receiving your revised manuscript.

Kind regards,

Godwin Upoki Anywar, BSc, Msc, PhD

Academic Editor

PLOS ONE

Reviewers' comments:

Reviewer's Responses to Questions

**Comments to the Author**

1. If the authors have adequately addressed your comments raised in a previous round of review and you feel that this manuscript is now acceptable for publication, you may indicate that here to bypass the “Comments to the Author” section, enter your conflict of interest statement in the “Confidential to Editor” section, and submit your "Accept" recommendation.

Reviewer #3: All comments have been addressed

Reviewer #4: All comments have been addressed

2. Is the manuscript technically sound, and do the data support the conclusions?

Reviewer #3: Partly

Reviewer #4: Yes

3. Has the statistical analysis been performed appropriately and rigorously? 

Reviewer #3: No

Reviewer #4: Yes

4. Have the authors made all data underlying the findings in their manuscript fully available?

Reviewer #3: Yes

Reviewer #4: Yes

5. Is the manuscript presented in an intelligible fashion and written in standard English?

Reviewer #3: No

Reviewer #4: Yes

6. Review Comments to the Author

Reviewer #3: The A. analysed a relevant number of woodcut images from the books of Historia Naturalis Brasiliae (1648). Though a comparison with visual and written sources they could identify a high number of taxa, and they used different tools for comparing data and check the taxonomical identification. In general, the topic of the study is original and very interesting, being an example of cultural and natural heritage that needs to be valorized.

Despite such great element of interest, I believe that the paper should be better organized for a contribution in an international scientific journal, since it deeply illustrates some findings of their work, but the topic is not introduced, clarified in the aims, and later discussed for a wider audience.

The introduction lacks referencing the recent international literature, where usually the aims of one paper originate (The selected references deal mostly with old historical records 4,20,21,11,13–19). Furthermore, the results of the previous works of the A. are not well introduced (Alcantara-Rodriguez M, Françozo M, Van Andel T. Looking into the flora of Dutch Brazil: botanical identifications of seventeenth century plant illustrations in the Libri Picturati. Sci Rep. 2021 Dec 1;11(1) and it is cited only in the methods!). Then, I understand that the previous one is a floristic paper, but it is a bit difficult deeply understand the novelty and originality of the paper.

In fact, the aims (analyze the correlations between the woodcuts and the other visual sources from Dutch Brazil, and trace back the remaining sources that were used to create the woodcuts) are not supported by critical analysis of the literature. Indeed, other scientific papers on such field of scientific Museology should be analyzed, considering other relevant cases studies. For example, when citing Hernandez, the A. should also cite the comments of G.B. Bettolo (Accademia Nazionale dei Lincei (Rome, Italy), to comment the Novae Hispaniae Thesaurus), and s relevant literature on such can be also found by the rich Portuguese scholars... (I add a few

Medeiros, Maria Franco Trindade, and Ulysses Paulino Albuquerque. "Food flora in 17th century northeast region of Brazil in Historia Naturalis Brasiliae." Journal of ethnobiology and ethnomedicine 10 (2014): 1-20.

Whitehead, P.J.P., 1976. The original drawings for the Historia naturalis Brasiliae of Piso and Marcgrave (1648). Journal of the Society for the Bibliography of Natural History, 7(4), pp.409-422.

da Cruz, A. and Praça, W.N., 2023. Reconnecting Knowledges: Historia Naturalis Brasiliae back to Indigenous Societies. In Toward an Intercultural Natural History of Brazil(pp. 142-165). Routledge.

Alsemgeest, A. and Bos, J., 2023. Census of the Copies of Willem Piso and Georg Marcgraf's Historia Naturalis Brasiliae (Leiden and Amsterdam: Elzevier, 1648). In Toward an Intercultural Natural History of Brazil: The Historia Naturalis Brasiliae Reconsidered (pp. 166-211). Routledge.

Ossenbach, C., 2017. Precursors of the Botanical Exploration of South America. Wilhelm Piso (1611-1678) and Georg Marcgrave (1610-1644). Lankesteriana, 17(1), pp.61-71.

Otherwise, if the topic of the paper is the image analysis the title should be changed, and different references should also be searched.

In the Methodological section the 4 classes are really ambiguous (the concept of similarity should be better explained...It means similar in the diagnostic elements??? .. and: which were the diagnostic elements? ... which were the limits of the classes of similarity??? In fact, the similarity can also arise from the habitus of a species, and different species can be sometime similar in the general characters, whereas the same species can appear different, if it is analysed in a different phenological status .... Then it must be clarified and specify the limit of each class! (very similar (woodcut and illustration share [almost] the same features), 2) moderately similar (they bear a great resemblance but do not share the same features), 3) slightly similar (woodcuts share some characteristics but not as many as in the previous category), and 4) different (not enough similar features between images to assume any correlation).

Furthermore, when using the world “correlation” (see the aims) I would expect some statistical analysis which can prove the level of correlation. Such part of analysis is completely missing... perhaps is it better avoiding such world.... The world “comparison” is much more appropriate.

I finally suggest explaining better the focus of the paper and rephrasing the title and some organization, adding an insertional literature on sources and methods more suitable. (I don’t enter too much in results and discussions, because the previous points are mandatory.

Reviewer #4: The manuscript Nature portrayed in images in Dutch Brazil: Tracing the sources of the plant woodcuts in the Historia Naturalis Brasiliae is an original contribution to the field. I only left minor corrections to the attached file.

7. PLOS authors have the option to publish the peer review history of their article (what does this mean?). If published, this will include your full peer review and any attached files.

Reviewer #3: No

Reviewer #4: No

---

## [Author Response · Author response to Decision Letter 1]

25 Nov 2023

Reviewer #3: The A. analysed a relevant number of woodcut images from the books of Historia Naturalis Brasiliae (1648). Though a comparison with visual and written sources they could identify a high number of taxa, and they used different tools for comparing data and check the taxonomical identification. In general, the topic of the study is original and very interesting, being an example of cultural and natural heritage that needs to be valorized.

Despite such great element of interest, I believe that the paper should be better organized for a contribution in an international scientific journal, since it deeply illustrates some findings of their work, but the topic is not introduced, clarified in the aims, and later discussed for a wider audience. 

Response: we appreciate the nice comments about our study. Indeed, it is crucial to value these natural history collections as rich biocultural heritage. Equally relevant is to generate new data out of them, that help us to understand the knowledge-making processes they entailed – in this case, the visual methods employed for the woodcuts. Our main aim was to trace back the sources used to elaborate the woodcuts in the HNB and IURNM, two different yet highly interconnected treatises. Other associated material also connected were the Libri Picturati, Marcgrave’s herbarium, and other scholars’ images. While many studies, especially those with an art-history approach, focused on the similarities you can see among images at first sight, due to common image elements, we used instead the botanical identifications of the plants portrayed. We have introduced our topic more explicitly in the introduction and we clarified the aims before discussing it later. 

Example in Lines 73 to 76: “The botanical imagery of these sources is the object of this study. By linking the materials and collections from Dutch Brazil, we aim to trace how these natural history books, like the HNB, were produced in the early modern period.”

Reviewer #3: The introduction lacks referencing the recent international literature, where usually the aims of one paper originate (The selected references deal mostly with old historical records 4,20,21,11,13–19).

Response: Indeed, our aim was highly inspired by modern discussions on the provenance of the woodcuts and by the lack of a systematic analysis of the woodcuts versus the Libri Picturati collection and other associated materials. The interconnections of these materials were clear to us when we identified the flora portrayed in the Theatrum, the Misc. Cleyeri and the Libri Principis. As pointed out by Whitehead and Boeseman (1989), Brienen (2006), and Scharf (2019), some of the illustrations in the Libri Picturati were similar in structure and composition to the woodcuts in the HNB. But, to what extent? Here is where our original botanical analysis played an important role. 

The historical records we used refer the scholars who analyzed the woodcuts or the visual material associated with the HNB when the collection was still in Berlin (e.g., Martius 1843, Lichtenstein, 1817, Schneider 1938). After Whitehead found it in the late 1970s in Krakow, these collections were analyzed by several scholars, mostly from an (art) historical point of view (e.g., Albertin, 1985; Whitehead and Boeseman, 1989; Boeseman, 1990; Teixeira, 1995; Brienen, 2006 Scharf, 2019, etc.). However, the reference numbers cited by the reviewer were disorganized because they were part of the first version of this manuscript (submitted in November 2022), which we revised and sent back in April 2023. The references in our revised manuscript (April 2023) correspond to modern and historical works on the matter under study. To further clarify the aim, we have added a more explicit explanation in the introduction and linked it in more detail to the literature. We now highlighted in more detail the works of these scholars from whom our work builds up and sometimes differs (as elaborated in our discussion). See the section “1.3 Behind the images: previous and present research” from lines 119 to 173.

Reviewer #3: Furthermore, the results of the previous works of the A. are not well introduced (Alcantara-Rodriguez M, Françozo M, Van Andel T. Looking into the flora of Dutch Brazil: botanical identifications of seventeenth century plant illustrations in the Libri Picturati. Sci Rep. 2021 Dec 1;11(1) and it is cited only in the methods!). Then, I understand that the previous one is a floristic paper, but it is a bit difficult deeply understand the novelty and originality of the paper.

In fact, the aims (analyze the correlations between the woodcuts and the other visual sources from Dutch Brazil, and trace back the remaining sources that were used to create the woodcuts) are not supported by critical analysis of the literature. Indeed, other scientific papers on such field of scientific Museology should be analyzed, considering other relevant cases studies. For example, when citing Hernandez, the A. should also cite the comments of G.B. Bettolo (Accademia Nazionale dei Lincei (Rome, Italy), to comment the Novae Hispaniae Thesaurus), and s relevant literature on such can be also found by the rich Portuguese scholars... (I add a few Medeiros, Maria Franco Trindade, and Ulysses Paulino Albuquerque. "Food flora in 17th century northeast region of Brazil in Historia Naturalis Brasiliae." Journal of ethnobiology and ethnomedicine 10 (2014): 1-20. 

Whitehead, P.J.P., 1976. The original drawings for the Historia naturalis Brasiliae of Piso and Marcgrave (1648). Journal of the Society for the Bibliography of Natural History, 7(4), pp.409-422. 

da Cruz, A. and Praça, W.N., 2023. Reconnecting Knowledges: Historia Naturalis Brasiliae back to Indigenous Societies. In Toward an Intercultural Natural History of Brazil (pp. 142-165). Routledge. 

Alsemgeest, A. and Bos, J., 2023. Census of the Copies of Willem Piso and Georg Marcgraf's Historia Naturalis Brasiliae (Leiden and Amsterdam: Elzevier, 1648). In Toward an Intercultural Natural History of Brazil: The Historia Naturalis Brasiliae Reconsidered (pp. 166-211). Routledge 

Ossenbach, C., 2017. Precursors of the Botanical Exploration of South America. Wilhelm Piso (1611-1678) and Georg Marcgrave (1610-1644). Lankesteriana, 17(1), pp.61-71. 

Otherwise, if the topic of the paper is the image analysis the title should be changed, and different references should also be searched. 

Response: We have clarified the paper's topic and added the recommended literature when appropriate. The works by Medeiros and Albuquerque (2014) and Ossenbach (2017) are not included in our revised version of this manuscript. The former compares 17th-century food plants as documented in the HNB with present food uses of these plants in Brazil, which has limited links to our paper. The latter focuses on two species of orchids described in the HNB and collected in Marcgrave’s herbarium. We have carefully examined and cited both works in our diachronic study on medicinal and other useful plants in the HNB and modern Brazil (Alcantara-Rodriguez et al., 2019). However, we decided to leave them out of this paper as they are unrelated to the provenance of the woodcuts in the HNB. 

The work by Hernández is cited as follows: “To navigate this vast corpus of literature, we first checked the HNB and IURNM for references to scholars who worked with tropical flora, such as Hernández (33), Monardes (34), or Clusius (35–37).” (lines 225-226). When the woodcuts did not match with the contemporary sources, we had to check in previous works because it was common at the time to “borrow” images from other treatises to complement the visual repertoire of texts on tropical flora. Furthermore, De Laet often cited those scholars, giving us clues as to whose works to study. Hence, instead of Bettolo (1990), we used the work of Almeida (2016) who systematized the entries of the scholars with the related plants, facilitating our search. As for Hernández, it is true that his oeuvre, and the corpus of work that originated from his expedition in New Spain, has profoundly impacted the cultural history of European-Native American knowledge exchanges and the history and historiography of early modern science – so much so that a proper comparison of the HNB and its trajectory to that of the Hernández corpus merits an entire new research project, or at least a dedicated paper to delve in detail into how these historical events and their cultural legacies converge and diverge. For this reason, we chose to remain close to the aim of our paper, that is, checking the possible sources for the woodcuts in the HNB. When analysing Hernández (1628), we did not find any correspondence with the woodcuts, otherwise, this fact would be reflected in our results and we would have discussed it and most likely facilitated our analysis with the work of Bettolo (1990, 1992).

We cite the work of Whitehead (1976) on the original drawings of the HNB (in which he discusses the Libri Picturati’s images as the possible source for the woodcuts and suspected it was somewhere in Poland) throughout the paper: see lines 114, 123, and 130.

In Methods (lines 181-184), we added Alsemgeest and Bos (2023) when we cited the digital copy of the HNB we used for this study (as there are over 300 copies around the world): “The surviving copies of the HNB amount to 302, distributed in collections and institutions worldwide (30). We used the digital-colored copy of the HNB located in Leiden University Library in the Netherlands [Shelfmark 1407 B3 (30)] (HNB Leiden Universiteitsbibliotheek) (Fig 1).

In addition, we incorporated the fantastic paper of Da Cruz and Neiva Praça (2023) in our discussion on the Indigenous Tupi-based plant names of the HNB and the Libri Picturati: “It is essential to notice that the Western scholars behind the HNB production oversimplified Indigenous and African languages under a few categories (such as “Tupi” or “Tupinambá,” a language belonging to the administrative and ecclesiastic lingua franca). Nevertheless, Indigenous groups spoke more than a thousand languages before the arrival of the Portuguese and their genocide, exposure to foreign diseases, and displacement (https://pib.socioambiental.org/en/Languages). Collaborative linguistic projects with modern Indigenous communities in Brazil, such as those by Da Cruz and Neiva Praça (70), highlighted the connections (i.e., cognates) in plant and animal names between the HNB and three Indigenous communities in Brazil. These studies have proven crucial to increasing access to these historical collections (often only available to the scholarly public) and recognizing the agency of those whose heritage may be connected to them”.

(lines 672 to 683). 

Moreover, we highlight a few more of these collaborative-based projects in the last part of the discussion (see Further research and recommendations).

Our paper on the botanical identifications of the Libri Picturari is cited in methods because we used the species identified in our previous work to cross-reference them with the plant species depicted in the woodcuts. In that way, we can see whether they are similar to each other (or not) and draw conclusions about their origins: “To compare woodcuts, vouchers, and plant illustrations, we cross-referenced all textual and visual sources for each plant species using our recent botanical identifications for the HNB, the IURNM, and the Libri Picturati (lines 201-203)”.

This manuscript does not compare to our previous one because the focus is not on identifying the plants depicted but on analyzing which sources were used to create the woodcuts in the natural history treatises under study (as the title states). However, we appreciate this remark, and we have now cited it in the introduction, in section “1.3. Behind the images: previous and present research previous research”. 

In lines 140 to 144 we added: “In our previous research, we identified the plant species depicted in the whole Brazilian collection within the Libri Picturati and found that a substantial number of species were documented and often illustrated in the HNB and IURNM (21). However, to what extent the images representing these species were comparable between the Libri Picturati and the HNB was still a mystery, as well as the potential authors behind their making”.

Reviewer #3: In the Methodological section the 4 classes are really ambiguous (the concept of similarity should be better explained...It means similar in the diagnostic elements??? .. and: which were the diagnostic elements? ... which were the limits of the classes of similarity??? In fact, the similarity can also arise from the habitus of a species, and different species can be sometime similar in the general characters, whereas the same species can appear different, if it is analysed in a different phenological status .... Then it must be clarified and specify the limit of each class! (very similar (woodcut and illustration share [almost] the same features), 2) moderately similar (they bear a great resemblance but do not share the same features), 3) slightly similar (woodcuts share some characteristics but not as many as in the previous category), and 4) different (not enough similar features between images to assume any correlation).

Response: We have now better elaborated these distinctions and the limits of each class in the text (Lines 211-221) “For the latter, we identified different degrees of similarities based on their botanical elements (or the lack thereof), spatial composition (indicating when the images were similar but reversed), shape, and form. We distinguished four categories: 1) very similar (woodcut and illustration share [almost] the same elements, shape, and form), 2) moderately similar (they bear a remarkable resemblance, especially in their morphology, but do not share precisely the same elements, or these are placed differently), 3) slightly similar (they are different in most elements yet some, such as the inflorescence, the fruits, a few leaves, etc., look-alike), and 4) different (the images do not present any similarity in their composition and botanical elements) ). The more similar a woodcut and its corresponding species in the Libri Picturati were, the more probably the woodcut was made after the image, or they both originated from the same source.”.

For example, referring to Samanea saman (an in-text example of a slightly similar woodcut compared to the oil painting in the Theatrum), we added in the results (Lines 598-600), "Since their images are only slightly similar, with similar inflorescences but different leaves and spatial composition, this weakens the possibility that they originated from the same source." 

The slightly similar images between the HNB and the Libri Picturati only accounted for 9%. After all, there are more differences between corresponding images between the woodcuts and the oil paintings (for the same species) than similarities. Hence, these illustrations later included in the Libri Picturati (or the sketches or drawings made before them) were not the primary sources used to elaborate the woodcuts in the HNB and IURNM (as suggested before). 

Finally, the reader can better visualize the differences between categories by looking at the illustrations compared to the woodcuts in the digital files we created (Appendices S2 and S4).

Reviewer #3: Furthermore, when using the word “correlation” (see the aims) I would expect some statistical analysis which can prove the level of correlation. Such part of analysis is completely missing... perhaps is it better avoiding such world.... The world “comparison” is much more appropriate.

Response: Although we have tabulated the data to calculate the percentages of each woodcut source (HNB/IURNM, Libri Picturati, Marcgrave’s herbarium, Eckhout/Post, and other sources – see Figs. 3 and 8 in the main manuscript), we have not used statistical tests because these require unambiguous quantitative differences, which do not apply to the comparison of drawings and therefore to our research. We used the term correlation by following the definition: “a mutual relationship or connection between two or more things” (Exford languages by Google) instead of its definition in statistics (“interdependence of variable quantities”, Oxford Languages). We do not assume a linear dependence of the variables but a connection between images that most likely share a common origin. Hence, to avoid ambiguity, we have followed the reviewer’s advice and changed the word correlation in the text for correspondence, matching, (inter)connections, (dis-) similarities, etc. 

Reviewer #3: I finally suggest explaining better the focus of the paper and rephrasing the title and some organization, adding an insertional literature on sources and methods more suitable. (I don’t enter too much in results and discussions, because the previous points are mandatory.

Response: The title of the paper relates to the aim of our research, the botanical-visual analysis we have conducted, and the context (“Dutch Brazil”). Therefore, we prefer to keep this title, also because it exactly reflects what we did in research: Tracing the sources of the woodcuts of the HNB (and the IURNM). We have done that by using our scientific identifications, comparing the images of the same plant species in the material remaining from Dutch Brazil, looking at other scholars/artists’ work, and quantifying our results to obtain a clear picture of the visual-making methods. 

With regard to the organization of the paper, we followed the reviewer's recommendations carefully and more explicitly mentioned the focus of the paper, edited the methods accordingly, and added the recommended literature when this was relevant to the scope of our manuscript. We hope we improved and enriched our paper by following these suggestions. 

Reviewer #4: The manuscript Nature portrayed in images in Dutch Brazil: Tracing the sources of the plant woodcuts in the Historia Naturalis Brasiliae is an original contribution to the field. I only left minor corrections to the attached file.

Response: We thank the reviewer for their feedback and for pointing out to us the minor mistakes in the text, which we have corrected.

---

## [Decision Letter · Decision Letter 2]

2 Jan 2024

PONE-D-22-26869R2Nature portrayed in images in Dutch Brazil: Tracing the sources of the plant woodcuts in the Historia Naturalis Brasiliae (1648)PLOS ONE

Dear Dr. Alcantara Rodriguez,

Thank you for submitting your manuscript to PLOS ONE. After careful consideration, we feel that it has merit but does not fully meet PLOS ONE’s publication criteria as it currently stands. Therefore, we invite you to submit a revised version of the manuscript that addresses the points raised during the review process.

We look forward to receiving your revised manuscript.

Kind regards,

Godwin Upoki Anywar, BSc, Msc, PhD

Academic Editor

PLOS ONE

Journal Requirements:

Reviewers' comments:

Reviewer's Responses to Questions

**Comments to the Author**

1. If the authors have adequately addressed your comments raised in a previous round of review and you feel that this manuscript is now acceptable for publication, you may indicate that here to bypass the “Comments to the Author” section, enter your conflict of interest statement in the “Confidential to Editor” section, and submit your "Accept" recommendation.

Reviewer #5: All comments have been addressed

Reviewer #6: (No Response)

2. Is the manuscript technically sound, and do the data support the conclusions?

Reviewer #5: Yes

Reviewer #6: Yes

3. Has the statistical analysis been performed appropriately and rigorously? 

Reviewer #5: N/A

Reviewer #6: N/A

4. Have the authors made all data underlying the findings in their manuscript fully available?

Reviewer #5: Yes

Reviewer #6: Yes

5. Is the manuscript presented in an intelligible fashion and written in standard English?

Reviewer #5: Yes

Reviewer #6: Yes

6. Review Comments to the Author

Reviewer #5: This is a comprehensive study on the provenance of woodcuts in Historia Naturalis Brasiliae, a historic collection of significant botanical and cultural value. Such interdisciplinary collaborations of authors from science and humanities are valuable and very much needed in the study of historic botanical collections. The manuscript is well written, methods are thorough and the arguments and conclusions are clear. The authors have carefully considered the comments addressed by the reviewers and provided substantial answers. I agree with the authors’ decision to keep the original title as it reflects the research they have carried out.

I provide below a few minor comments to help amend the text (line numbers refer to the final text without track changes):

Line 92: Alternatively: “included 26 plant images (15 drawings and 11 proof woodcuts)”, for the sake of clarity as later a reference to 26 images is made (in line 563).

Line 127: "which woodcut" replace with "the woodcut of which"

Line 212: “degrees of similarity” instead of “degrees of similarities”.

Line 449: I would suggest that the authors don’t use the term “Photoshop”. It directly refers to a software developed by a private company and there might be legal restrictions in the use of it. “Image manipulation” could be an alternative.

Lines 531-532: I suggest another phrasing here: When connecting the images and descriptions in De Laet’s manuscript to the corresponding species and woodcuts in the HNB (Supporting Information S5), we find 388 plant descriptions and 165 plant entries with the word Icon next to the entry. Out of these… etc.

Line 641: It is not clear what “their” refers to in heading 4.1, and perhaps it is redundant, please consider omitting it or rephrasing the heading.

669-673: “Our database with all… natural history collections”: This sentence better fits as a conclusion, please consider transferring it to the conclusions section.

Line 697: Strictly speaking, Gessner was not an author of a popular herbal. His only botanical book, Historia Plantarum, was not finished before his death and was published only in 2001. The authors could rather refer to Fuchs and Matthioli who both wrote popular herbals. And either add birth and death dates for both scholars or for none.

Line 732: Not only images never seen before but perhaps also “images of plant species never seen before”? At least not seen in the European scholar circles. This also adds to the argument of generating a new visual language.

Lines 746-7: Consider an alternative phrasing, such as: “In contrast to what is argued in previous research”, the Theatrum … etc

Reviewer #6: The manuscript presents a systematic analysis of the woodcut images from mid-seventeenth century, namely images of natural elements that originated in Dutch Brazil and circulated in Europe. These images played a significant role in disseminating scientific botanical knowledge. To analyze the (dis-) similarities among the historical visual sources, the authors built a database in FileMaker Pro with all woodcuts and illustrations organized by species and created a spreadsheet with the background information. The analysis are very interesting and the approach is rater creative. I find this work quite interesting overall and well written. With the corrections and edits in response to the previous reviewers in my opinion the work is suitable for publication the way it is.

7. PLOS authors have the option to publish the peer review history of their article (what does this mean?). If published, this will include your full peer review and any attached files.

Reviewer #5: No

Reviewer #6: No

---

## [Author Response · Author response to Decision Letter 2]

11 Jan 2024

Journal Requirements

Response: We carefully reviewed the reference list and edited all the references according to the journal guidelines (see changes in “Manuscript with track changes”). We eliminated number 72 (Kusukawa, 2010) because we are not using Gessner’s visual methods anymore in comparison to those used in the HNB). We adjusted the referencing numbers that followed. Lastly, we are not including any retracted papers. 

Review Comments to the Author

Reviewer #5: This is a comprehensive study on the provenance of woodcuts in Historia Naturalis Brasiliae, a historic collection of significant botanical and cultural value. Such interdisciplinary collaborations of authors from science and humanities are valuable and very much needed in the study of historic botanical collections. The manuscript is well written, methods are thorough and the arguments and conclusions are clear. The authors have carefully considered the comments addressed by the reviewers and provided substantial answers. I agree with the authors’ decision to keep the original title as it reflects the research they have carried out.

I provide below a few minor comments to help amend the text (line numbers refer to the final text without track changes):

Line 92: Alternatively: “included 26 plant images (15 drawings and 11 proof woodcuts)”, for the sake of clarity as later a reference to 26 images is made (in line 563).

Line 127: "which woodcut" replace with "the woodcut of which"

Line 212: “degrees of similarity” instead of “degrees of similarities”.

Response: We agreed with the reviewer’s comments and edited the text according to those.

Reviewer #5: Line 449: I would suggest that the authors don’t use the term “Photoshop”. It directly refers to a software developed by a private company and there might be legal restrictions in the use of it. “Image manipulation” could be an alternative.

Response: To avoid any potential problem with this trademarked term, we used instead Image editing, as similarly advised by the reviewer.

Reviewer #5: Lines 531-532: I suggest another phrasing here: When connecting the images and descriptions in De Laet’s manuscript to the corresponding species and woodcuts in the HNB (Supporting Information S5), we find 388 plant descriptions and 165 plant entries with the word Icon next to the entry. Out of these… etc.

Response: We agreed with the reviewer’s phrasing and we have applied it to the text.

Reviewer #5: Line 641: It is not clear what “their” refers to in heading 4.1, and perhaps it is redundant, please consider omitting it or rephrasing the heading.

Response: We changed the pronoun to “its”, as it refers to Brazilian flora. We left the term “Tupi names” as we aim to highlight that the vernacular Indigenous names (in this case from the Tupi linguistic branch) were crucial when organizing the images. 

Reviewer #5: 669-673: “Our database with all… natural history collections”: This sentence better fits as a conclusion, please consider transferring it to the conclusions section.

Response: We completely agreed with the reviewer, so we have moved this sentence to the conclusion (lines 805-809).

Reviewer #5: Line 697: Strictly speaking, Gessner was not an author of a popular herbal. His only botanical book, Historia Plantarum, was not finished before his death and was published only in 2001. The authors could rather refer to Fuchs and Matthioli who both wrote popular herbals. And either add birth and death dates for both scholars or for none.´

Response: We meant to write “natural history books” instead of herbals as we were referring to Gessner’s Historia Plantarum, instead of his postmortem herbal. However, in this context, it is better to refer to popular herbals as those written by Fuchs and Matthioli. In contrast to Marcgrave, the authors could review and edit their work, including the images. As suggested by the reviewer, we added the dates of birth and death for both authors to be consistent.

Reviewer #5: Line 732: Not only images never seen before but perhaps also “images of plant species never seen before”? At least not seen in the European scholar circles. This also adds to the argument of generating a new visual language.

Lines 746-7: Consider an alternative phrasing, such as: “In contrast to what is argued in previous research”, the Theatrum … etc.

Response: We agreed with the reviewer and adjusted the phrasing accordingly in the text.

Reviewer #6: The manuscript presents a systematic analysis of the woodcut images from mid-seventeenth century, namely images of natural elements that originated in Dutch Brazil and circulated in Europe. These images played a significant role in disseminating scientific botanical knowledge. To analyze the (dis-) similarities among the historical visual sources, the authors built a database in FileMaker Pro with all woodcuts and illustrations organized by species and created a spreadsheet with the background information. The analysis are very interesting and the approach is rater creative. I find this work quite interesting overall and well written. With the corrections and edits in response to the previous reviewers in my opinion the work is suitable for publication the way it is.

Response: We kindly thank the reviewer for appreciating our work.

---

## [Editor Report · Decision Letter 3]

12 Feb 2024

Nature portrayed in images in Dutch Brazil: Tracing the sources of the plant woodcuts in the Historia Naturalis Brasiliae (1648)

PONE-D-22-26869R3

Dear Dr. Alcantara Rodriguez,

We’re pleased to inform you that your manuscript has been judged scientifically suitable for publication and will be formally accepted for publication once it meets all outstanding technical requirements.

Kind regards,

Godwin Upoki Anywar, BSc, Msc, PhD

Academic Editor

PLOS ONE
---

## [Editor Report · Acceptance letter]

23 Feb 2024

PONE-D-22-26869R3 

PLOS ONE

Dear Dr. Alcantara-Rodriguez, 

I'm pleased to inform you that your manuscript has been deemed suitable for publication in PLOS ONE. Congratulations! Your manuscript is now being handed over to our production team.

Kind regards, 

on behalf of

Dr. Godwin Upoki Anywar 

Academic Editor

PLOS ONE